# LC–MS Lipidomics: Exploiting a Simple High-Throughput Method for the Comprehensive Extraction of Lipids in a Ruminant Fat Dose-Response Study

**DOI:** 10.3390/metabo10070296

**Published:** 2020-07-17

**Authors:** Benjamin Jenkins, Martin Ronis, Albert Koulman

**Affiliations:** 1NIHR BRC Core Metabolomics and Lipidomics Laboratory, University of Cambridge, Pathology Building Level 4, Addenbrooke’s Hospital, Cambridge CB2 0QQ, UK; bjj25@medschl.cam.ac.uk; 2College of Medicine, Department of Pharmacology & Experimental Therapeutics, Louisiana State University Health Sciences Centre, 1901 Perdido Str., New Orleans, LA 70112, USA; mronis@lsuhsc.edu

**Keywords:** odd chain lipids, lipid profiling, Folch, protein precipitation, sample preparation, relative lipid composition (Mol%)

## Abstract

Typical lipidomics methods incorporate a liquid–liquid extraction with LC–MS quantitation; however, the classic sample extraction methods are not high-throughput and do not perform well at extracting the full range of lipids especially, the relatively polar species (e.g., acyl-carnitines and glycosphingolipids). In this manuscript, we present a novel sample extraction protocol, which produces a single phase supernatant suitable for high-throughput applications that offers greater performance in extracting lipids across the full spectrum of species. We applied this lipidomics pipeline to a ruminant fat dose–response study to initially compare and validate the different extraction protocols but also to investigate complex lipid biomarkers of ruminant fat intake (adjoining onto simple odd chain fatty acid correlations). We have found 100 lipids species with a strong correlation with ruminant fat intake. This novel sample extraction along with the LC–MS pipeline have shown to be sensitive, robust and hugely informative (>450 lipids species semi-quantified): with a sample preparation throughput of over 100 tissue samples per day and an estimated ~1000 biological fluid samples per day. Thus, this work facilitating both the epidemiological involvement of ruminant fat, research into odd chain lipids and also streamlining the field of lipidomics (both by sample preparation methods and data presentation).

## 1. Introduction

Lipids are generally understood as a class of molecules that have a high solubility in organic solvents and typically contain or originate from fatty acids. Although, lipids may be commonly derived, research has shown that there is a huge variety both structurally and functionally (potentially >40,000 [1]); where they play a vital role in energy production and storage [2,3], regulation and signalling [4,5], provide structure and support and membrane formation [6]. Lipids are now emerging as biomarkers of dietary/nutritional intakes [7] as well as indicators of pathophysiological status [8,9,10,11]. As a reaction of lipid-pathophysiological involvements, the field of lipidomics has emerged as a discipline that examines and quantifies a large proportion of the lipids present in a given sample set.

Lipidomics requires an effective isolation protocol that comprehensively extracts lipids from the sample as well as an analytical method that allows their identification and quantitation. The typical analyte isolation protocols (with/without minor adaptations) that are often used in the literature include three different liquid–liquid extractions: Folch and colleagues [12] (cited >65,000 times), Bligh and Dyer [13] (cited >52,000 times) or Matyash and colleagues [14] (cited >1000 times). Although these extraction protocols are heavily cited and do result in adequate results, there are several caveats with their use. Firstly, there is the need to perform duplicate extraction in non-fluid samples to ensure optimal recovery of the lipid analytes. This is extremely time consuming, especially for the Folch and the Bligh and Dyer methods. Secondly, there are reasonable concerns that using a biphasic extraction (producing immiscible aqueous and organic phases) may result in a loss of relatively polar lipids (e.g., acyl-carnitines and gangliosides) into the disposed aqueous fractions (consisting of mostly methanol and water in these extraction protocols: Folch and the Bligh and Dyer). There are publications that use a single phase extraction protocol but they do not appear to solve the problem of extracting the relatively more polar lipids since a mixture of methanol, chloroform and tert-butyl methyl ether were used [15,16].

The technique overwhelmingly used tor analysing the lipidome is mass spectrometry hyphenated with chromatography (LC–MS) due to its sensitivity and selectivity; furthermore by using a high-resolution accurate mass instrument (e.g., Orbitrap or Time-of-Flight instruments), a huge number of analytes can be analysed simultaneously. Reversed phase chromatography is the predominant chromatographic technique employed to separate the analytes before entering the mass spectrometer to determine their structure and concentration. Variants of a liquid chromatography method utilising a C18-column with a water and acetonitrile mix for the weak eluting mobile phase and acetonitrile and propan-2-ol for the strong eluting mobile phase are the most commonly used [17,18,19,20,21,22]. These reversed phase C18-column methods both separate lipid based on their lipid class assignment (i.e., either phosphatidylcholines or phosphatidyethanolamines head group) and their fatty-acyl composition (i.e., chain length and degree of unsaturation) with some degree of isomeric separation.

In this study, we present a lipidomics pipeline that including a novel analyte isolation protocol utilising a single phase, which results in a comprehensive lipid extraction suitable for a full range of lipid polarities (from polar to non-polar lipids species). This lipidomics method was tested, validated and then applied in a rat model investigating ruminant fat biomarkers via a beef tallow dose response dietary investigation.

## 2. Results

This lipidomics LC–MS method incorporating both of the described sample preparation protocols: protein precipitation (chloroform: methanol: acetone, ~7:3:4) and Folch liquid–liquid (chloroform: methanol: water, ~7:3:4), were utilised for the quantitation of lipids in liver samples from Sprague–Dawley rats who received one of four experimental diets overfed at 17% above matched growth.

A comparison of the two sample preparation methods on the extraction of the stable isotope-labelled internal standards are shown in the figure below (see Figure 1). A comparison on the samples’ endogenous individual lipid classes are shown in the Appendix A.

As shown the intensity of seven of the internal standards are statistically significantly higher in the protein precipitation protocol when compared to the Folch liquid–liquid protocol (between ~30% to ~2500% higher), whereas, only two of the internal standards were higher in the Folch liquid–liquid protocol (between ~20% and ~37% higher). Additionally, there is far less variation in the protein precipitation liquid extraction protocol: 10 out of 13 internal standards had ~2% to ~72% less variation in their coefficient of variation (CV). The protein precipitation liquid extraction protocol has also been shown to produce a significantly higher detection of the samples’ endogenous lipids, both producing a higher total number of lipid detected (Folch-LLE: 455 lipid species, PPLE: 472 lipid species) and a statistically significantly higher total intensity for twelve of the sixteen lipid classes detected (see Appendix A). Taken as a whole, the protein precipitation protocol (chloroform: methanol: acetone, ~7:3:4) showed a greater extraction capability across the full lipid hydro-philicity/phobicity range and across the internal standards. Additionally, the high throughput of the protein precipitation protocol allows for over 100 tissue samples to be extracted per day (including dissection, weighing and tissue extraction ready for LC–MS analysis), whereas, the Folch liquid–liquid protocol could take up to three- to four-times longer due to the necessity of duplicate extractions and the delicacy of liquid–liquid phase separation. The throughput of the protein precipitation protocol (chloroform: methanol: acetone, ~7:3:4) on fluid samples allows an estimated ~1000 biological fluid sample extractions per day (including aliquoting, sample extraction ready for LC–MS analysis) when utilising basic laboratory fluid handling equipment/robots (throughput data not shown here).

The liver lipid concentration (nM/mg) for each experimental diet group of rats are shown in the table below (see Table 1), along with the correlation (trendline equation, slope significance, R^2^ and successive change across the groups) of the measured lipid concentration with the percentage composition of ruminant fat (beef tallow) in each experimental diet. An R^2^ threshold of 0.75, slope significance *p*-value < 0.05 and successive increase/decrease were set to establish if there was a strong correlation between the lipid concentration and the ruminant fat composition.

## 3. Discussion

Out of 472 lipids detected and semi-quantified, 100 showed a strong relationship with the dietary intake of ruminant fat with 35 species increasing and 65 species decreasing as the percentage of ruminant fat in the diet increased (NB. ruminant fat as a percentage composition with corn-oil and medium chain triacylglyceride oil). Interestingly, ceramides generally increased, whilst cardiolipins, sphingomyelins and triacylglycerides generally decreased as the dietary composition of ruminant fat rose. According to the literature, a rise in liver ceramides is typically associated with aggravated non-alcoholic fatty liver disease (NAFLD) and insulin resistance [23], this in conjunction with a decrease in cardiolipins (which are indicative of mitochondrial remodelling and dysfunction [24]) may suggest that the changes in the experimental diets here are detrimental for these pathologies. However, there was a clear decrease in the triacylglycerides (particularly evident in the unsaturated odd chain triacylglycerides), which is explicitly representative of an ameliorated pathology [25]. As previously published [26], many NAFLD and insulin resistance factors were mitigated as the ruminant fat increased in these diets, including: a reduction in the total body weight (g), total fat mass (%), serum ALT (U/mL) and degree of steatosis determined by Oil Red O staining, notably, the inflammatory marker TNFα did not change significantly (trend: *p*-value = 0.52). A key characteristic of NAFLD development is the accumulation of hepatic triacylglycerides [27]; therefore, the data here suggests that these dietary changes may be beneficial for NAFLD and insulin resistance by aiding in a reduced hepatic triacylglyceride load: possible mechanisms here include a lower saturated fatty acid composition resulting in a lower fatty acid incorporation into hepatic triacylglycerides and/or a higher pass-through of the medium chain triglyceride oil directly into the mitochondria stimulation fatty acid metabolism [26]. Work presented by Gonzalez-Cantero and colleagues [28] showed that hepatic triacylglyceride content were correlated with insulin resistance and these relationships were independently to the inflammatory marker TNFα. Therefore, it appears that the hepatic triacylglyceride load may be paramount in the development of NAFLD and insulin resistance, which is supported in the literature [29].

According to the literature, odd chain fatty acids are considered biomarkers of their dietary intake and particularly accredited as a biomarkers of ruminant fat intake (e.g., milk, butter and beef tallow, etc.); however, there is a vast amount of conflicting data [2]. Some studies have shown both positive correlations (either individual odd chain lipids or total odd chain lipids) and some studies have shown there were no significant correlations. Although these studies may conflict in their findings they all present their data as relative compositions (Mol%), which is the typical way lipid data appear in the literature [30]. By expressing the lipid data as relative compositions (Mol%), it normalises the data to the total fat in that sample; however, presenting the lipid data in this way confounds the results by interconnecting the individual data points. This interconnection can cause false positive and/or false negative conclusions (type 1 and 2 errors), i.e., if a single lipid increases it will artificially decrease the other(s) due to the Mol% calculation. As shown in the figure below (see Figure 2), the concentration of the total lipids containing either even chain or odd chain fatty acids and a combination are shown. Although lipids containing odd chain fatty acids did increase, albeit not statistically significantly; *p*-value: 0.197, it was also not proportionate to the increase in dietary ruminant fat. As shown, the lipids containing even chain fatty acids did significantly inversely decrease (slope *p*-value: 0.0189) as the dietary ruminant fat increased. Interestingly, due to both the decrease in the even chain lipids and the consistency of the odd chain lipids, if the relative composition (Mol%) of the lipids were calculated, there appears to be a statistically significant increase in the odd chain lipids (see Figure 3); however, this is an artefact of changes in the even chain lipids and a consequence of interconnecting the data.

## 4. Materials and Methods

### 4.1. Chemicals and Standards

Stable isotope-labelled internal standards purchased from Sigma Aldrich (Haverhill, Suffolk, UK) include: N-palmitoyl-d31-D-erythro-sphingosine (abbreviated to IS_Cer_16:0-d31); order number: 868516P, 1-palmitoyl-d31-2-oleoyl-sn-glycero-3-phosphate (abbreviated to IS_PA_34:1-d31); order number: 860453P, 1-palmitoyl-d31-2-oleoyl-sn-glycero-3-phosphocholine (abbreviated to IS_PC_34:1-d31); order number: 860399P, 1-palmitoyl-d31-2-oleoyl-sn-glycero-3-phosphoethanolamine (abbreviated to IS_PE_34:1-d31); order number: 860374P, 1-palmitoyl-d31-2-oleoyl-sn-glycero-3-[phospho-rac-(1-glycerol)] (abbreviated to IS_PG_34:1-d31); order number: 860384P, 1-palmitoyl-d31-2-oleoyl-sn-glycero-3-phosphoinositol (abbreviated to IS_PI_34:1-d31); order number: 860042P, 1,2-dimyristoyl-d54-sn-glycero-3-[phospho-L-serine] (abbreviated to IS_PS_28:0-d54); order number: 860401P, N-palmitoyl-d31-D-erythro-sphingosylphosphorylcholine (abbreviated to IS_SM_34:1-d31); order number: 868584P. Stable isotope-labelled internal standards purchased from QMX Laboratories Ltd. (QMX Laboratories Ltd., Thaxted, Essex, UK) include: Heptadecanoic-d33 acid (abbreviated to IS_FA_17:0-d33); order number: D-5261, N-tetradecylphosphocholine-d42 (abbreviated to IS_LPC_14:0-d42); order number: D-5885, Glyceryl tri(pentadecanoate-d29) (abbreviated to IS_TG_45:0-d87); order number: D-5265, Butyryl-d7-L-carnitine (abbreviated to IS_Car_4:0-d7); order number: D-7761, Hexadecanoyl-L-carnitine-d3 (abbreviated to IS_Car_16:0-d3); order number: D-6646.

Quality control standards (LIPID-QC) purchased from Cayman Chemical Company (Cambridge Bioscience, Cambridge, UK) include: Lysophosphatidylcholines (egg); order number: 24331, Phosphotidylcholines (egg); order number: 24343, Lysophosphatidylethanolamines (egg); order number: 25844, Phosphatidylethanolamines (bovine); order number: 16878, Phosphotidlethanolamine (soy); order number: 25845, Lysophosphatidyinositols (porcine liver); order number: 26016, Phosphatidylserines (soy); order number: 25847, Ceramides mixture; order number: 22853, Ceramides (non-hydroxy); order number: 24833, Ceramides (hydroxy); order number: 24834, Sphingomyelins (from bovine spinal cord); order number: 22674, Sphingomylins (egg); order number: 24345, Phosphatidylglycerols (egg); order number: 25846, Phosphatidic acid (egg); order number: 24344, Sulfatides (bovine); order number: 24323, Purified mixed gangliosides (bovine); order number: 24856, TLC Neutral Glycosphingolipid Mixture (bovine and porcine); order number: 1505, 2-Palmitoyl Glycerol; order number: CAY17882, 1,2-Dipalmitoyl-sn-glycerol; order number: CAY10008648. Quality control standards purchased from Sigma Aldrich include: Soy PC (95%); order number: 441601G, C18(Plasm)-18:1-PC; order number: 852467C, Brain CPE; order number: 860066P, Liver PI; order number: 840042P, Brain lyso PS; order number: 850092P, Milk SM Sphingomyelin (Milk, Bovine); order number: 860063P, Galactocerebrosides from bovine brain; order number: C4905, Glucosylceramide (Soy); order number: 131304P, Triglyceride mix, C2–C10; order number: 17810-1amp-s, Fish oil from menhaden; order number: F8020, Anhydrous butter fat, Cardiolipin solution from bovine heart; order number: C1649, Brain PI(4)P; order number: 840045P.

Commercially available blank human serum was purchased from BioIVT (Royston, Hertfordshire, UK; order number: HUMANSRMPNN. All solvents and additives were of HPLC grade or higher and purchased from Sigma Aldrich unless otherwise stated.

LIPID-IS: the lipid stable isotope-labelled internal standard was prepared by dissolving each of the individual lipid standards into chloroform: methanol (1:1) solution to produce a 1 mM primary stock solution. From each of these stock solutions, 1 mL was transferred into a volumetric flask and diluted with methanol to reach a final working solution concentration of 5 µM in methanol of IS_Cer_16:0-d31, IS_FA_17:0-d33, IS_LPC_14:0-d42, IS_PA_34:1-d31, IS_PC_34:1-d31, IS_PE_34:1-d31, IS_PG_34:1-d31, IS_PI_34:1-d31, IS_PS_28:0-d54, IS_SM_34:1-d31, IS_TG_45:0-d87.

ACYL-CARNITINE-IS: the acyl-carnitine stable isotope-labelled internal standard was prepared by dissolving each powdered stock into methanol to achieve a 5 mM stock solution. Taking 1 mL of the IS_Car_4:0-d7 and IS_Car_16:0-d3 stock solutions and diluting these into methanol until a final working solution of 5 µM was achieved for IS_Car_4:0-d7 and IS_Car_16:0-d3.

LIPID-QC: the lipid quality control standards were prepared by diluting each lipid mix to achieve a 50 µg/mL working stock solution in propan-2-ol: acetonitrile: water (2:1:1, respectively).

### 4.2. Extraction

Lipids were isolated comparing two methods; firstly, a novel protein-precipitation liquid extraction and secondly the liquid–liquid extraction previously described by Folch and colleagues [12] in an adapted version as we described previously [31]. Tissue quantities ranged from ~2–50 mg and fluid samples from 10–50 µL (e.g., plasma/serum) were tested (data not shown here).

#### 4.2.1. Protein Precipitation Liquid Extraction Protocol (PPLE)

The protein-precipitation liquid extraction protocol was as follows: the tissue samples were weighed (NB. fluid samples were pipetted) and transferred into a 2 mL screw cap Eppendorf plastic tube (Eppendorf, Stevenage, UK) along with a single 5 mm stainless steel ball bearing. Immediately, 400 µL of chloroform: methanol (2:1, respectively) solution was added to each sample, followed by thorough mixing. The samples were then homogenised in the chloroform: methanol (2:1, respectively) using a Bioprep 24-1004 homogenizer (Allsheng, Hangzhou, China) run at speed; 4.5 m/s, time; 30 s for 2 cycles. Then, 400 µL of chloroform, 100 µL of the LIPID-IS (5 µM in methanol) and 100 µL of the CARNITINE-IS (5 µM in methanol) was added to each sample. The samples were homogenised again using a Bioprep 24-1004 homogenizer run at speed; 4.5 m/s, time; 30 s for 2 cycles. To ensure fibrous material was diminished, the samples were sonicated for 30 min in a water bath sonicator (Advantage-Lab, Menen, Belgium). Then, 400 µL of acetone was added to each sample. The samples were thoroughly vortexed and centrifuged for 10 min at ~20,000× *g* to pellet any insoluble material at the bottom of the vial. The single layer supernatant was pipetted into separate 2 mL screw cap amber-glass auto-sampler vials (Agilent Technologies, Cheadle, UK); being careful not to break up the solid pellet at the bottom of the tube. The organic extracts (chloroform, methanol, acetone composition, ~1.4 mL) were dried down to dryness using a Concentrator Plus system (Eppendorf, Stevenage, UK) run for 60 min at 60 °C. The samples were reconstituted in 100 µL of 2:1:1 (propan-2-ol, acetonitrile and water, respectively) then thoroughly vortex. The reconstituted sample was transferred into a 250 μL low-volume vial insert inside a 2 mL amber glass auto-sample vial ready for liquid chromatography with mass spectrometry detection (LC–MS) analysis.

#### 4.2.2. Folch Liquid–Liquid Extraction Protocol (Folch LLE)

The Folch liquid–liquid extraction protocol is as follows: the tissue samples were weighed (NB. fluid samples were pipetted) and transferred into a 2 mL screw cap Eppendorf plastic tube (Eppendorf, Stevenage, UK) along with a single 5 mm stainless steel ball bearing. Immediately, 400 µL of chloroform: methanol (2:1, respectively) solution was added to each sample, followed by thorough mixing. The samples were then homogenised in the chloroform: methanol (2:1, respectively) using a Bioprep 24-1004 homogenizer (Allsheng, Hangzhou, China) run at speed; 4.5 m/s, time; 30 s for 2 cycles. Then, 400 µL of chloroform, 100 µL of the LIPID-IS (5 µM in methanol) and 100 µL of the ACYL-CARNITINE-IS (5 µM in methanol) was added to each sample. The samples were homogenised again using a Bioprep 24-1004 homogenizer run at speed; 4.5 m/s, time; 30 s for 2 cycles. To ensure fibrous material was diminished, the samples were sonicated for 30 min in a water bath sonicator. Then, 400 µL of HPLC water was added to each samples. The samples were thoroughly vortexed and centrifuged for 10 min at ~20,000 g to separate the two immiscible fractions. The organic fractions (the lower layer, mostly chloroform; ~700 µL) and aqueous fractions (the upper layer, methanol and water; ~700 µL) were pipetted into separate 2 mL screw cap amber-glass auto-sampler vials (Agilent Technologies, Cheadle, UK); being careful not to break up the solid pellet between the layers. To ensure complete lipid isolation a double extraction protocol was followed; 1 mL of chloroform: methanol (2:1, respectively) solution was added to each sample, along with 400 µL of HPLC water. The samples were thoroughly vortexed and centrifuged for 10 min at ~20,000× *g*. The organic fractions and aqueous fractions were pipetted into the corresponding 2 mL screw cap amber-glass auto-sampler vials containing the initial extracts (again being careful not to break up the solid pellet between the layers). The combined organic extracts (~1.4 mL) were dried down to dryness using a Concentrator Plus system (Eppendorf, Stevenage, UK) run for 60 min at 60 °C. The samples were reconstituted in 100 µL of 2:1:1 (propan-2-ol, acetonitrile and water, respectively) then thoroughly vortex. The reconstituted sample was transferred into a 250 μL low-volume vial insert inside a 2 mL amber glass auto-sample vial ready for liquid chromatography with mass spectrometry detection (LC–MS) lipidomics analysis.

### 4.3. LC–MS Method

Full chromatographic separation of intact lipids was achieved using a Shimadzu HPLC System (Shimadzu UK Limited, Milton Keynes, UK) with the injection of 10 µL onto a Waters Acquity UPLC^®^ CSH C18 column (Waters, Hertfordshire, UK); 1.7 µm, I.D. 2.1 mm × 50 mm, maintained at 55 °C. Mobile phase A was 6:4, acetonitrile and water with 10 mM ammonium formate. Mobile phase B was 9:1, propan-2-ol and acetonitrile with 10 mM ammonium formate. The flow was maintained at 500 µL per minute through the following gradient: 0.00 min_40% mobile phase B; 0.40 min_43% mobile phase B; 0.45 min_50% mobile phase B; 2.40 min_54% mobile phase B; 2.45 min_70% mobile phase B; 7.00 min_99% mobile phase B; 8.00 min_99% mobile phase B; 8.3 min_40% mobile phase B; 10 min_40% mobile phase B. The sample injection needle was washed using 9:1, 2-propan-2-ol and acetonitrile. The mass spectrometer used was the Thermo Scientific Exactive Orbitrap with a heated electrospray ionisation source (Thermo Fisher Scientific, Hemel Hempstead, UK). The mass spectrometer was calibrated immediately before sample analysis using positive and negative ionisation calibration solution (recommended by Thermo Scientific). Additionally, the heated electrospray ionisation source was optimised at 50:50 mobile phase A to mobile phase B for spray stability (capillary temperature; 300 °C, source heater temperature; 420 °C, sheath gas flow; 40 (arbitrary), auxiliary gas flow; 15 (arbitrary), spare gas; 3 (arbitrary), source voltage; 4 kV. The mass spectrometer scan rate set at 4 Hz, giving a resolution of 25,000 (at 200 *m*/*z*) with a full-scan range of *m*/*z* 100 to 1800 with continuous switching between positive and negative mode. 

### 4.4. Data Processing

Thermo Xcalibur Quan Browser (Thermo Fisher Scientific, Hemel Hempstead, UK) data processing involved the integration of the internal standard extracted ion chromatogram (EIC) peaks at the expected retention times (see Table 2). The EIC were selected from the ionisation mode for each analyte class; the ionisation mode is dependent on the molecular chemistry of the analytes, i.e., basic chemical groups ordinarily result in positive ionisation (e.g., [M+H]^+^, M+H-H_2_O]^+^, [M+Na]^+^, [M+NH_4_]^+^, [M+K]^+^) whereas acidic chemical groups typically result in negative ionisation (e.g., [M-H]^−^).

As shown in the table above (see Table 2), the internal standards have multiple ionisation products, these are the result of numerous ionisation mechanism (for example IS_TG_45:0-d87 having different adducts: [M+H]^+^, [M+Na]^+^, [M+K]^+^ and [M+NH_4_]^+^, present) as well as an isotopic distribution (e.g., IS_TG_45:0-d87 having either the expected eighty-seven or fewer deuterium atoms present) all reasonably expected ions were included into the EIC for each internal standard (see Figure 4).

The adduct composition of the total EIC produced from each of the ionisation mechanisms are shown in the figure below (see Figure 5).

The data processing also involved the integration of the individual lipid (and derivatives) species at their expected retention time (see Appendix A) allowing for a maximum of ±0.1 min of retention time drift: any retention time drift greater than ±0.1 min resulted in the exclusion of the analyte leading to a ‘Not Found’ result (i.e., zero concentration). A list of the analyte classes along with the number of species detected within each class are shown in the table below (see Table 3). The expected adducts for each analyte class and the internal standard used for semi-quantitation are also shown.

The lipid quality control (QC) standards were analysed with each batch of samples, these QC standards were used to check the retention times for the analytes ensuring that isobaric analytes were separated and expected analyte retention times remained robust.

Through the Thermo Xcalibur Quan Browser software, the responses of the analytes were normalised to the relevant internal standard response (producing area ratios) (see Appendix A), these area ratios corrected the intensity for any extraction and instrument variations. The area ratios were then blank corrected where intensities less than three times the blank samples were set to a ‘Not Found’ result (i.e., zero concentration). The accepted area ratios were then multiplied by the concentration of the internal standard to give the analyte concentrations. The results for fluid samples were expressed in molar concentrations (typically µM or nM). For tissue samples, the calculated concentrations of the analytes were then divided by the amount of tissue (in mg) used in the extraction protocol to give the final results in µM per mg of tissue extracted (µM/mg).

### 4.5. Animal Intervention

Sprague–Dawley rats (Harlan, IN, USA) were overfed using one of four experimental diets (n = 6–9 per group) at 17% above matched growth via an intragastric cannula surgically inserted as previously described [26]. Animals had ad libitum access to water throughout the experiments. The four experimental diets were 70% fat (% energy) including different amounts of medium chain triacylglycerides oil (MCT), beef tallow and corn oil; the fat composition of each diet are shown in the table below (see Table 4).

Protein (19% whey protein), vitamin and mineral contents were the same in all diets. Diets were formulated to meet the caloric and nutritional recommendations established by the National Research Council (NRC), but were fed at a level that exceeded the recommended caloric intake by 17% to increase weight gain and adiposity and produce steatohepatitis.

Liver tissue was collected after 21 days. All experimental procedures were ethically approved by the Institutional Animal Care and Use Committee at the University of Arkansas for Medical Science.

## 5. Conclusions

This lipidomics protocol has been developed to quantify lipids across a broad range of hydrophobicities, from acyl-carnitines through to long chain glycerolipids. The extraction method produces a single liquid supernatant phase ideal for high-throughput workflows with an increased extraction capability over the frequently published liquid–liquid extraction previously published by Folch and colleagues [12].

Following the establishment and validation of this method, we applied it to a ruminant fat dose response dietary intervention in Sprague–Dawley rats, where we found 100 lipid species correlated strongly with the composition of ruminant fat within the diet.

It has been previously suggested that dietary ruminant fat is beneficial/protective in type 2 diabetes [32], the results presented in this manuscript suggest possible target mechanisms that need to be examined could include ceramide fatty acid compositions, cardiolipin remodeling, sphingomyelins and/or triacylglycerides concentration (particularly unsaturated odd chain species) and their associated fatty acid compositions, as well as the liver total lipid content.

## Figures and Tables

**Figure 1 metabolites-10-00296-f001:**
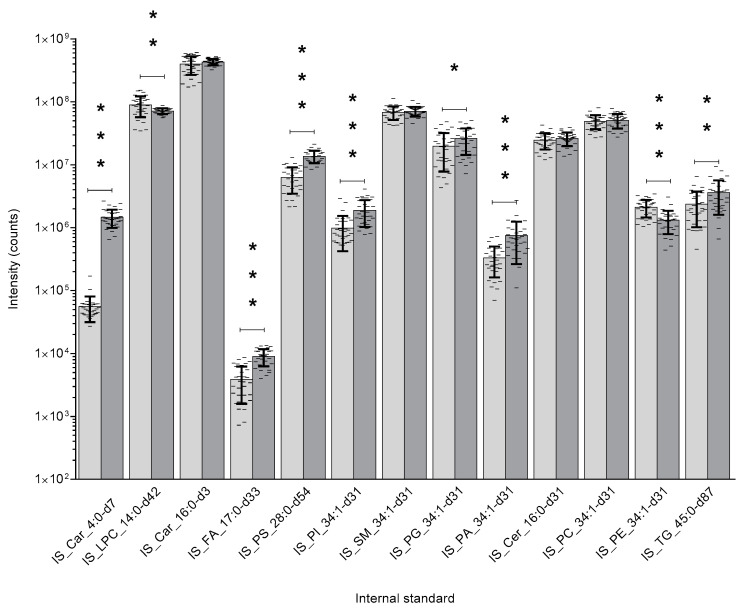
This figure shows the comparison between the two lipid extraction techniques regarding their extraction efficiency of the stable isotope internal standards from the rat liver samples (Folch liquid–liquid extraction: chloroform: methanol: water, ~7:3:4 
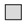
, and Protein precipitation liquid extraction: chloroform: methanol: acetone, ~7:3:4 
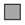
). n = 34 rat liver samples per extraction method. The intensity of the internal standards were measured by liquid chromatography with mass spectrometry. The significance of the difference between the two extraction protocols are shown by the p-value star system; where *p* ≤ 0.05 was considered statistically significant (* *p* < 0.05, ** *p* < 0.01, *** *p* < 0.001). Error bars represent ± standard deviation. Lipid internal standard include: Butyryl-d7-L-carnitine (abbreviated to IS_Car_4:0-d7), N-tetradecylphosphocholine-d42 (abbreviated to IS_LPC_14:0-d42), hexadecanoyl-L-carnitine-d3 (abbreviated to IS_Car_16:0-d3), heptadecanoic-d33 acid (abbreviated to IS_FA_17:0-d33), 1,2-dimyristoyl-d54-sn-glycero-3-[phospho-L-serine] (abbreviated to IS_PS_28:0-d54), 1-palmitoyl-d31-2-oleoyl-sn-glycero-3-phosphoinositol (abbreviated to IS_PI_34:1-d31), N-palmitoyl-d31-D-erythro-sphingosylphosphorylcholine (abbreviated to IS_SM_34:1-d31), 1-palmitoyl-d31-2-oleoyl-sn-glycero-3-[phospho-rac-(1-glycerol)] (abbreviated to IS_PG_34:1-d31), 1-palmitoyl-d31-2-oleoyl-sn-glycero-3-phosphate (abbreviated to IS_PA_34:1-d31), N-palmitoyl-d31-D-erythro-sphingosine (abbreviated to IS_Cer_16:0-d31), 1-palmitoyl-d31-2-oleoyl-sn-glycero-3-phosphocholine (abbreviated to IS_PC_34:1-d31), 1-palmitoyl-d31-2-oleoyl-sn-glycero-3-phosphoethanolamine (abbreviated to IS_PE_34:1-d31), glyceryl tri(pentadecanoate-d29) (abbreviated to IS_TG_45:0-d87).

**Figure 2 metabolites-10-00296-f002:**
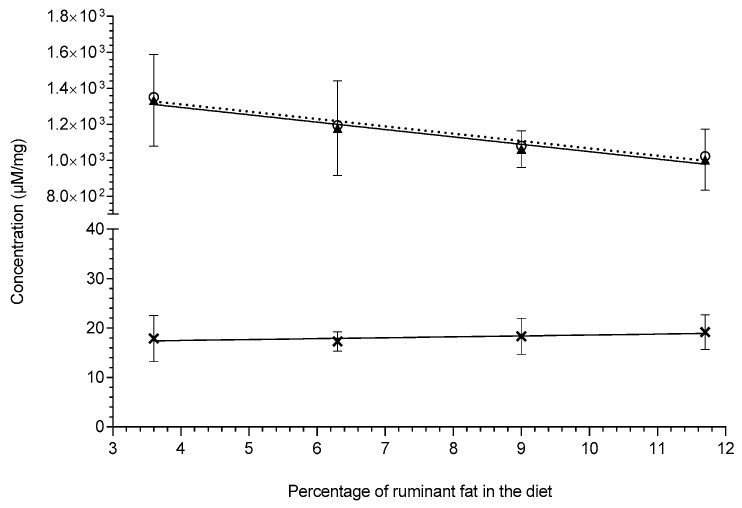
This figure shows the change in the liver lipid concentrations across the four high-fat diets fed to Sprague–Dawley rats (n = 8–9 per group): total odd chain lipids (symbol: **X**, trendline: 
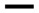
, gradient: 0.183 ± 0.0962, R^2^ = 0.64, slope significance *p*-value: 0.197); total even chain lipids (symbol: 
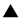
, trendline: 
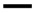
, gradient: −41.0 ± 5.71, R^2^ = 0.963, slope significance *p*-value: 0.0189); total lipids containing both even and of odd chain (symbol: **O**, trendline: 
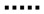
, gradient: −40.8 ± 5.79, R^2^ = 0.961, slope significance *p*-value: 0.0195. Lipid concentrations (µM/mg) are shown as means ± standard deviation and were extracted via the protein precipitation liquid extraction protocol (chloroform: methanol: acetone, ~7:3:4).

**Figure 3 metabolites-10-00296-f003:**
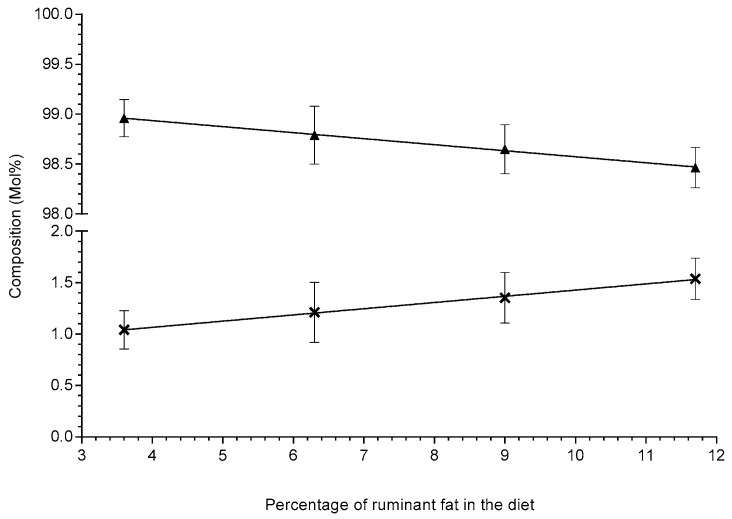
This figure shows the relative compositional (Mol%) change in the total odd chain lipids (symbol: **X**, trendline: 
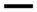
, gradient: 0.0604 ± 0.00203, R^2^ = 0.998, slope significance *p*-value: 0.0011) and the total even chain lipids (symbol: 
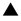
, trendline: 
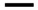
, gradient: −0.0604 ± 0.00203, R^2^ = 0.998, slope significance *p*-value: 0.0011) across the four high-fat diets in Sprague–Dawley rats (n = 8–9 per group). Lipid compositions (Mol%) are shown as means ± standard deviation and were extracted via the protein precipitation liquid extraction protocol (chloroform: methanol: acetone, ~7:3:4). Diet one: 3.6% beef tallow; diet two: 6.3% beef tallow; diet three: 9.0% beef tallow; diet four: 11.7% beef tallow.

**Figure 4 metabolites-10-00296-f004:**
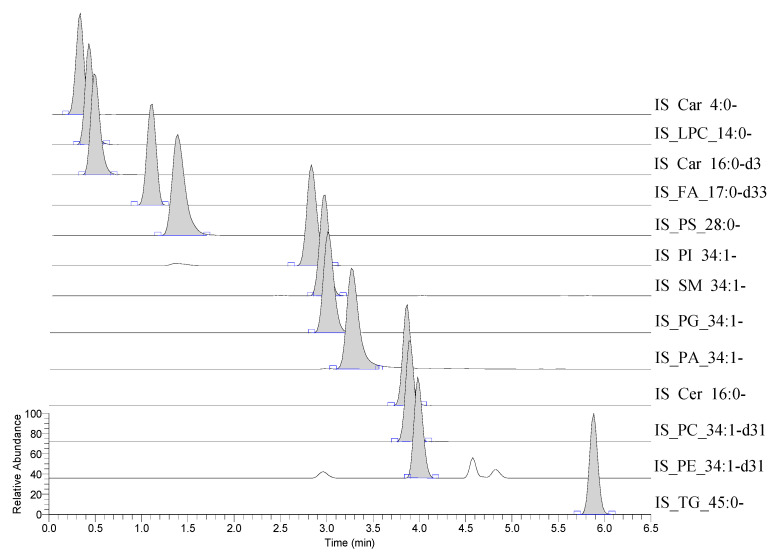
This figure shows a spiked (5 µM in methanol) commercial human plasma extracted ion chromatogram (EIC) for the stable isotope-labelled internal standards (lipids and acyl-carnitines): butyryl-d7-L-carnitine (abbreviated to IS_Car_4:0-d7); area ~5.9 × 10^6^ counts, N-tetradecylphosphocholine-d42 (abbreviated to IS_LPC_14:0-d42); are ~2.1 × 10^8^ counts, hexadecanoyl-L-carnitine-d3 (abbreviated to IS_Car_16:0-d3); area ~5.8 × 10^8^ counts, heptadecanoic-d33 acid (abbreviated to IS_FA_17:0-d33); area ~1.1 × 10^4^ counts, 1,2-dimyristoyl-d54-sn-glycero-3-[phospho-L-serine] (abbreviated to IS_PS_28:0-d54); area ~2.2 × 10^7^ counts, 1-palmitoyl-d31-2-oleoyl-sn-glycero-3-phosphoinositol (abbreviated to IS_PI_34:1-d31): area ~1.1 × 10^7^ counts, N-palmitoyl-d31-D-erythro-sphingosylphosphorylcholine (abbreviated to IS_SM_34:1-d31); 1.4 × 10^8^ counts, 1-palmitoyl-d31-2-oleoyl-sn-glycero-3-[phospho-rac-(1-glycerol)] (abbreviated to IS_PG_34:1-d31); area ~6.1 × 10^7^ counts, 1-palmitoyl-d31-2-oleoyl-sn-glycero-3-phosphate (abbreviated to IS_PA_34:1-d31); area ~1.3 × 10^7^ counts, N-palmitoyl-d31-D-erythro-sphingosine (abbreviated to IS_Cer_16:0-d31); area ~2.2 × 10^8^ counts, 1-palmitoyl-d31-2-oleoyl-sn-glycero-3-phosphocholine (abbreviated to IS_PC_34:1-d31); area ~2.5 × 10^8^ counts, 1-palmitoyl-d31-2-oleoyl-sn-glycero-3-phosphoethanolamine (abbreviated to IS_PE_34:1-d31); area ~6.3 × 10^7^ counts, glyceryl tri(pentadecanoate-d29) (abbreviated to IS_TG_45:0-d87); area ~3.2 × 10^7^ counts.

**Figure 5 metabolites-10-00296-f005:**
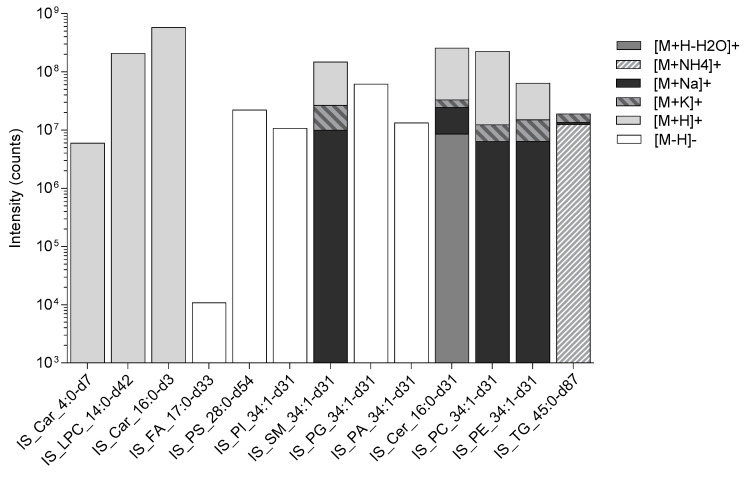
This figure shows the intensity of the extracted ion chromatogram for each stable isotope-labelled internal standard along with the ionisation adduct composition: [M+H]^+^, [M+H-H_2_O]^+^, [M+NH_4_]^+^, [M+Na]^+^, [M+K]^+^ and [M-H]^−^. Butyryl-d7-L-carnitine (abbreviated to IS_Car_4:0-d7), N-tetradecylphosphocholine-d42 (abbreviated to IS_LPC_14:0-d42), hexadecanoyl-L-carnitine-d3 (abbreviated to IS_Car_16:0-d3), heptadecanoic-d33 acid (abbreviated to IS_FA_17:0-d33), 1,2-dimyristoyl-d54-sn-glycero-3-[phospho-L-serine] (abbreviated to IS_PS_28:0-d54), 1-palmitoyl-d31-2-oleoyl-sn-glycero-3-phosphoinositol (abbreviated to IS_PI_34:1-d31), N-palmitoyl-d31-D-erythro-sphingosylphosphorylcholine (abbreviated to IS_SM_34:1-d31), 1-palmitoyl-d31-2-oleoyl-sn-glycero-3-[phospho-rac-(1-glycerol)] (abbreviated to IS_PG_34:1-d31), 1-palmitoyl-d31-2-oleoyl-sn-glycero-3-phosphate (abbreviated to IS_PA_34:1-d31), N-palmitoyl-d31-D-erythro-sphingosine (abbreviated to IS_Cer_16:0-d31), 1-palmitoyl-d31-2-oleoyl-sn-glycero-3-phosphocholine (abbreviated to IS_PC_34:1-d31), 1-palmitoyl-d31-2-oleoyl-sn-glycero-3-phosphoethanolamine (abbreviated to IS_PE_34:1-d31), glyceryl tri(pentadecanoate-d29) (abbreviated to IS_TG_45:0-d87).

**Table 1 metabolites-10-00296-t001:** This table shows the liver lipid concentrations from the Sprague–Dawley rats who received one of four experimental diets (n = 8–9 per group). Diet 1: 50% corn oil, 16.4% MCT oil and 3.6% beef tallow; Diet 2: 35% corn oil, 28.7% MCT oil and 6.3% beef tallow; Diet 3: 20% corn oil, 41.0% MCT oil and 9.0% beef tallow; Diet 4: 5% corn oil, 53.3% MCT oil and 11.7% beef tallow. MCT: medium chain triglyceride oil. Lipid are shown in their shorthand notations with the number of carbons and unsaturated bonds in the fatty acid moiety separated by a colon; acyl-carnitines (Carn), ceramides (Cer), cardiolipins (CL), diacylglycerols (DG), gangliosides (GM1), hexosylceramides (Hex-Cer), lyso-phosphatidylcholines (LPC), lyso-phosphatidyethanolamines (LPE), lyso-phosphatidylinositols (LPI), lyso-cardiolipins (Lyso_CL), phosphatidic acids (PA), phosphatidylcholines (PC), phosphatidylethanolamines (PE), phosphatidylglycerol (PG), phosphatidylinositols (PI), phosphatidylserines (PS), sulfatides (S), sphingomyelins (SM), triacylglycerides (TG). Lipid concentrations (nM/mg) are shown as mean ± standard deviation and were extracted via the protein precipitation liquid extraction protocol (chloroform: methanol: acetone, ~7:3:4). Correlation between the ruminant fat composition of the diet and the lipid concentration are depicted by the trendline equation (* denotes statistical significance of the slope: *p*-value < 0.05) and coefficient of determination (R^2^). Lipid concentrations continually increasing/decreasing across the groups as the ruminant fat composition of the diet increase are emphasised in the ‘Successive change across group’ column. Lipids with a significant slope (*p*-value < 0.05) with an R^2^ greater than 0.75 and successively increasing/decreasing are in bold and highlighted.

	Diet 1	Diet 2	Diet 3	Diet 4	Trendline Equation	R^2^	Successive Change Across Groups
Carn_(C00:0)	99,100 ± 19,200	90,200 ± 38,800	126,000 ± 29,000	92,400 ± 49,600	y = 581x + 97,500	0.02	
Carn_(C02:0)	10,400 ± 4150	10,700 ± 3280	12,900 ± 6740	11,700 ± 5300	y = 226x + 9700	0.49	
**Carn_(C03:0)**	**3430 ± 1380**	**3590 ± 1820**	**3760 ± 2170**	**3830 ± 2240**	**y = 50.7x + 3260 ***	**0.97**	**Increasing**
Carn_(C03:0-2COOH)	713 ± 326	436 ± 299	943 ± 438	700 ± 604	y = 17.3x + 565	0.08	
Carn_(C03:0-OH)	ND ± ND	1.46 ± 4.39	2.1 ± 5.94	6.16 ± 13	y = 0.87x − 4.59	0.85	
Carn_(C03:1)	108 ± 64.2	105 ± 58.2	101 ± 75.6	75.5 ± 35.3	y = −3.76x + 126	0.78	Decreasing
Carn_(C04:0)	1040 ± 847	675 ± 342	1560 ± 659	793 ± 309	y = 5.33x + 976	0.00	
Carn_(C04:0-OH)	952 ± 429	702 ± 503	1730 ± 944	1230 ± 619	y = 69x + 626	0.30	
Carn_(C04:1)	134 ± 56	102 ± 94.4	89.6 ± 63.3	104 ± 78.4	y = −3.79x + 136	0.49	
Carn_(C05:0)	490 ± 534	359 ± 173	857 ± 577	553 ± 437	y = 25.4x + 370	0.18	
Carn_(C05:1)	319 ± 125	255 ± 87.3	317 ± 205	227 ± 99.5	y = −7.93x + 340	0.36	
Carn_(C06:0)	324 ± 287	183 ± 61.5	635 ± 296	234 ± 123	y = 6.74x + 292	0.01	
Carn_(C06:0-2COOH)	511 ± 145	362 ± 124	742 ± 246	477 ± 213	y = 10.3x + 444	0.05	
Carn_(C08:0)	168 ± 174	107 ± 66.6	412 ± 185	186 ± 215	y = 13.3x + 117	0.12	
Carn_(C08:1)	81.7 ± 50.9	102 ± 56.5	124 ± 54.8	88.3 ± 62.7	y = 1.55x + 87.2	0.08	
Carn_(C10:0-2COOH)	353 ± 178	361 ± 124	378 ± 146	296 ± 177	y = −5.7x + 391	0.31	
Carn_(C12:0)	0.265 ± 0.339	0.147 ± 0.353	0.495 ± 0.291	0.412 ± 0.402	y = 0.0292x + 0.106	0.43	
Carn_(C14:0)	11.1 ± 6.52	9.39 ± 8.48	14.8 ± 5.36	13 ± 5.69	y = 0.411x + 8.92	0.38	
Carn_(C15:0)	4.33 ± 2.64	3.76 ± 2.87	4.88 ± 1.52	3.65 ± 0.927	y = −0.0341x + 4.42	0.04	
Carn_(C16:0)	618 ± 339	457 ± 315	739 ± 411	462 ± 226	y = −6.89x + 622	0.03	
Carn_(C16:0-OH)	1.2 ± 1.56	1.52 ± 1.98	1.61 ± 1.35	2.46 ± 2.18	y = 0.143x + 0.601	0.86	Increasing
Carn_(C16:2)	9.89 ± 7.09	9.69 ± 4.49	7.61 ± 4.01	10.9 ± 4.92	y = 0.0352x + 9.25	0.01	
Carn_(C17:0)	19.2 ± 8.93	13.9 ± 6.7	22.3 ± 9.04	18.4 ± 7.41	y = 0.222x + 16.8	0.05	
Carn_(C18:0)	627 ± 220	541 ± 170	656 ± 154	620 ± 221	y = 3.48x + 584	0.06	
Carn_(C18:0-OH)	5.47 ± 3.31	6.8 ± 3.5	5.41 ± 4.18	8.14 ± 4.52	y = 0.245x + 4.58	0.44	
Carn_(C18:1)	1530 ± 1090	1230 ± 931	1900 ± 958	1130 ± 449	y = −19.6x + 1600	0.04	
Carn_(C18:2)	824 ± 754	631 ± 525	374 ± 157	505 ± 259	y = −45x + 927	0.67	
Carn_(C18:3)	16.8 ± 16.4	14.7 ± 13.3	11.5 ± 6.38	14.8 ± 5.16	y = −0.341x + 17.1	0.29	
Carn_(C20:0)	43.1 ± 38.9	41.9 ± 28.2	32.5 ± 20.4	45.9 ± 22.9	y = −0.037x + 41.1	0.00	
**Carn_(C22:5)**	**2.6 ± 3.13**	**2.34 ± 2.81**	**1.7 ± 1.25**	**1.66 ± 1.06**	**y = −0.128x + 3.06 ***	**0.91**	**Decreasing**
Cer_(32:1)	5.17 ± 1.26	4.31 ± 0.783	5.34 ± 1.78	5.4 ± 1.75	y = 0.0637x + 4.57	0.19	
**Cer_(33:1)**	**2.29 ± 1.92**	**1.68 ± 1.26**	**1.48 ± 0.544**	**1.27 ± 0.925**	**y = −0.121x + 2.6 ***	**0.92**	**Decreasing**
Cer_(34:0)	0.918 ± 0.57	1.19 ± 1.13	2.53 ± 2.19	1.93 ± 1.51	y = 0.162x + 0.402	0.60	
Cer_(34:1)	1.49 ± 0.581	1.23 ± 0.881	1.79 ± 1.01	2.56 ± 0.989	y = 0.14x + 0.699	0.71	
Cer_(35:0)	ND ± ND	ND ± ND	0.0674 ± 0.191	ND ± ND			
**Cer_(35:1)**	**1.03 ± 1.06**	**1.61 ± 1.06**	**1.63 ± 1.08**	**2.16 ± 1.45**	**y = 0.126x + 0.641 ***	**0.91**	**Increasing**
**Cer_(36:0)**	**7.26 ± 2.79**	**9.98 ± 5.3**	**10.5 ± 4.07**	**13.1 ± 2.89**	**y = 0.668x + 5.1 ***	**0.95**	**Increasing**
Cer_(36:1)	54.5 ± 26.2	55.4 ± 10.4	59.7 ± 16.3	71.7 ± 24.5	y = 2.07x + 44.5	0.83	Increasing
Cer_(36:2)	6.59 ± 3.55	5.23 ± 1.18	5.94 ± 2.23	6.81 ± 3.13	y = 0.0507x + 5.75	0.06	
Cer_(37:1)	2.23 ± 0.767	1.9 ± 0.723	2.31 ± 0.772	4.23 ± 1.09	y = 0.237x + 0.851	0.61	
Cer_(37:2)	ND ± ND	ND ± ND	0.066 ± 0.187	ND ± ND			
Cer_(38:0)	2.14 ± 1.19	3.82 ± 3.78	2.1 ± 0.715	2.9 ± 0.626	y = 0.0207x + 2.58	0.01	
Cer_(38:1)	47.9 ± 18.8	34.1 ± 7.21	39.5 ± 10.3	56 ± 13	y = 1.1x + 36	0.16	
Cer_(38:2)	6.76 ± 3.87	4.76 ± 1.57	4.7 ± 0.967	8.02 ± 1.98	y = 0.138x + 5.01	0.09	
Cer_(39:0)	0.188 ± 0.0814	0.221 ± 0.149	0.243 ± 0.153	0.724 ± 0.454	y = 0.0604x − 0.118	0.68	Increasing
Cer_(39:1)	10.5 ± 3.44	8.36 ± 1.6	10.6 ± 1.62	20.3 ± 3.17	y = 1.17x + 3.48	0.58	
Cer_(39:2)	0.436 ± 0.447	0.332 ± 0.333	0.416 ± 0.342	1.65 ± 0.755	y = 0.138x − 0.347	0.58	
**Cer_(40:0)**	**3.71 ± 0.85**	**4.12 ± 0.933**	**4.74 ± 1.44**	**6.17 ± 0.98**	**y = 0.296x + 2.42 ***	**0.92**	**Increasing**
Cer_(40:1)	126 ± 37.9	116 ± 18.7	146 ± 51	227 ± 39.1	y = 12.3x + 59.4	0.73	
Cer_(40:2)	65.5 ± 24.7	50.8 ± 8.2	63 ± 13.4	105 ± 12.7	y = 4.84x + 34	0.52	
**Cer_(41:0)**	**2.95 ± 0.547**	**3.91 ± 1.03**	**4.94 ± 1.2**	**7.37 ± 1.36**	**y = 0.529x + 0.744 ***	**0.94**	**Increasing**
**Cer_(41:1)**	**72.1 ± 18.6**	**89.7 ± 20.4**	**130 ± 88.3**	**210 ± 48.4**	**y = 16.8x − 3.18 ***	**0.91**	**Increasing**
Cer_(41:2)	51.4 ± 19.1	49.6 ± 11.4	63.3 ± 6.75	144 ± 22.5	y = 10.8x − 5.52	0.70	
**Cer_(42:0)**	**6.21 ± 2.01**	**8.11 ± 2.54**	**11.5 ± 5.17**	**12.2 ± 3.76**	**y = 0.791x + 3.45 ***	**0.95**	**Increasing**
**Cer_(42:1)**	**310 ± 63.3**	**444 ± 130**	**624 ± 539**	**773 ± 305**	**y = 58.1x + 93.2 ***	**1.00**	**Increasing**
Cer_(42:2)	575 ± 161	599 ± 87.4	721 ± 120	1180 ± 187	y = 71.7x + 220	0.79	Increasing
Cer_(42:3)	105 ± 40.2	92 ± 19.5	103 ± 26.1	151 ± 16.5	y = 5.52x + 70.5	0.54	
**Cer_(43:0)**	**1.31 ± 0.425**	**2.05 ± 0.699**	**2.43 ± 0.806**	**3.07 ± 0.982**	**y = 0.21x + 0.611 ***	**0.99**	**Increasing**
**Cer_(43:1)**	**131 ± 31.2**	**218 ± 62.9**	**253 ± 87.4**	**396 ± 111**	**y = 30.7x + 14.3 ***	**0.94**	**Increasing**
Cer_(43:2)	42.9 ± 7.62	58.9 ± 11.9	64.2 ± 12.8	115 ± 17.2	y = 8.21x + 7.46	0.84	Increasing
**Cer_(44:1)**	**16.7 ± 4.31**	**28.5 ± 8.34**	**38.9 ± 26.1**	**43 ± 14.8**	**y = 3.31x + 6.47 ***	**0.96**	**Increasing**
**Cer_(44:2)**	**0.908 ± 0.271**	**1.19 ± 0.288**	**1.72 ± 1.08**	**2.08 ± 0.644**	**y = 0.15x + 0.328 ***	**0.99**	**Increasing**
Cer_(45:1)	0.46 ± 0.215	1.19 ± 0.531	1.32 ± 0.618	1.58 ± 0.746	y = 0.129x + 0.149	0.88	Increasing
Cer_(45:2)	ND ± ND	0.00348 ± 0.0104	0.00503 ± 0.0142	ND ± ND	y = 0.000574x − 0.000137	1.00	
**Cer_(46:1)**	**0.0023 ± 0.0069**	**0.0192 ± 0.0266**	**0.0409 ± 0.0568**	**0.0568 ± 0.0824**	**y = 0.00686x − 0.0227 ***	**1.00**	**Increasing**
Cer_(46:2)	0.83 ± 0.187	1.53 ± 0.439	1.57 ± 0.516	1.79 ± 0.679	y = 0.108x + 0.603	0.82	Increasing
**CL_(66:02)**	**8.65 ± 6.02**	**22.3 ± 17.1**	**32.3 ± 17.1**	**37.7 ± 31.3**	**y = 3.6x − 2.29 ***	**0.97**	**Increasing**
CL_(66:03)	15.8 ± 10.1	23.6 ± 16.4	24.2 ± 13.2	17.3 ± 8.8	y = 0.189x + 18.8	0.02	
CL_(66:04)	15.5 ± 10.7	21.8 ± 10.4	29.7 ± 16.7	27 ± 15.3	y = 1.57x + 11.5	0.76	
**CL_(66:05)**	**40.5 ± 20**	**39.1 ± 10.7**	**18.5 ± 10.3**	**1 ± 0.911**	**y = −5.15x + 64.2 ***	**0.92**	**Decreasing**
CL_(66:06)	66.1 ± 37	67.9 ± 26.4	65.1 ± 38.1	22.6 ± 13.2	y = −4.94x + 93.2	0.62	
CL_(67:02)	4.6 ± 5.88	15.8 ± 13.3	11.3 ± 6.02	10.1 ± 10	y = 0.444x + 7.05	0.11	
CL_(67:03)	13.4 ± 9.85	22.4 ± 14	14.1 ± 6.36	5.97 ± 4.69	y = −1.13x + 22.6	0.35	
**CL_(67:05)**	**0.848 ± 1.74**	**1.4 ± 2.57**	**2.58 ± 2.02**	**3.16 ± 1.62**	**y = 0.301x − 0.303 ***	**0.98**	**Increasing**
CL_(68:00)	0.484 ± 0.502	0.898 ± 1.1	0.998 ± 1.12	0.519 ± 1.01	y = 0.00759x + 0.667	0.01	
CL_(68:01)	105 ± 91.1	260 ± 222	190 ± 107	142 ± 163	y = 1.52x + 163	0.01	
CL_(68:02)	469 ± 385	1010 ± 799	664 ± 355	434 ± 480	y = −16.7x + 772	0.05	
CL_(68:03)	631 ± 421	1030 ± 670	560 ± 261	237 ± 199	y = −61.2x + 1080	0.43	
CL_(68:04)	367 ± 179	439 ± 197	294 ± 113	184 ± 112	y = −25.7x + 518	0.68	
CL_(69:04)	40.4 ± 31.9	66.9 ± 41.7	56.3 ± 27.3	28.8 ± 18.6	y = −1.68x + 61	0.12	
CL_(69:05)	64.8 ± 37.5	67.7 ± 29.7	51.2 ± 20.4	31.1 ± 15.2	y = −4.36x + 87	0.83	
CL_(69:06)	49.7 ± 32.3	49.7 ± 20.5	29.3 ± 11	25.1 ± 10.8	y = −3.49x + 65.1	0.86	
CL_(69:07)	0.245 ± 0.736	1.06 ± 0.942	1.33 ± 1.17	0.685 ± 0.79	y = 0.0589x + 0.38	0.19	
CL_(70:01)	1.45 ± 1.87	6.38 ± 7.09	4.32 ± 2.94	3.91 ± 5.53	y = 0.197x + 2.51	0.12	
CL_(70:02)	61.2 ± 59.7	102 ± 90.1	54.6 ± 31.2	33.4 ± 40.7	y = −4.84x + 99.9	0.35	
CL_(70:03)	470 ± 379	796 ± 600	559 ± 271	340 ± 303	y = −23.2x + 719	0.18	
CL_(70:04)	1270 ± 814	2010 ± 1300	1510 ± 656	950 ± 761	y = −54.1x + 1850	0.18	
CL_(70:05)	1770 ± 889	2210 ± 1060	1670 ± 641	1200 ± 780	y = −83.3x + 2350	0.49	
CL_(70:06)	1280 ± 629	1570 ± 595	1560 ± 688	1440 ± 897	y = 17.4x + 1330	0.20	
CL_(70:07)	882 ± 480	1110 ± 399	1210 ± 657	971 ± 551	y = 13.6x + 939	0.11	
**CL_(70:08)**	**256 ± 199**	**245 ± 108**	**147 ± 56.7**	**89.7 ± 52.6**	**y = −22.1x + 354 ***	**0.93**	**Decreasing**
CL_(70:09)	25.8 ± 18.3	27.1 ± 13.6	17.7 ± 8.62	14.1 ± 7.15	y = −1.65x + 33.8	0.83	
CL_(71:02)	0.744 ± 1.36	2.59 ± 3	1.41 ± 1.38	0.451 ± 0.714	y = −0.0763x + 1.88	0.08	
CL_(71:03)	7.35 ± 7.45	20.2 ± 18.2	18.2 ± 11	9.9 ± 8.38	y = 0.209x + 12.3	0.01	
CL_(71:04)	19.5 ± 12.7	51.1 ± 39.5	65.7 ± 41.9	43 ± 33	y = 3.15x + 20.7	0.32	
CL_(71:05)	56.9 ± 35	116 ± 76.6	130 ± 56.9	107 ± 69.8	y = 6.09x + 55.9	0.44	
CL_(71:06)	106 ± 65.4	164 ± 78.1	192 ± 74.8	164 ± 88.4	y = 7.48x + 99.3	0.52	
CL_(71:07)	76.3 ± 55.8	104 ± 37.6	120 ± 42.8	101 ± 54.8	y = 3.34x + 74.8	0.41	
**CL_(71:08)**	**23.9 ± 25.3**	**17.1 ± 6.59**	**5.98 ± 4.09**	**0.44 ± 0.928**	**y = −3.02x + 34.9 ***	**0.98**	**Decreasing**
CL_(72:01)	0.251 ± 0.441	1.36 ± 1.4	0.758 ± 0.648	0.825 ± 1.56	y = 0.0415x + 0.481	0.10	
CL_(72:02)	0.874 ± 1.29	10.3 ± 12.3	4.1 ± 2.73	4.24 ± 7.52	y = 0.144x + 3.77	0.02	
CL_(72:03)	22.6 ± 21.7	55.2 ± 54.7	23.4 ± 12.2	17.9 ± 25	y = −1.7x + 42.8	0.12	
CL_(72:04)	246 ± 202	374 ± 291	228 ± 110	237 ± 229	y = −6.41x + 320	0.11	
CL_(72:05)	2310 ± 1300	3190 ± 1750	2650 ± 1080	2000 ± 1500	y = −54.4x + 2950	0.14	
CL_(72:06)	10,800 ± 5640	12,200 ± 4950	9770 ± 3660	5820 ± 3810	y = −643x + 14,600	0.67	
**CL_(72:07)**	**28,500 ± 15,500**	**25,400 ± 6760**	**17,100 ± 6180**	**7210 ± 4430**	**y = −2670x + 40,000 ***	**0.96**	**Decreasing**
**CL_(72:08)**	**31,100 ± 17,500**	**24,200 ± 5090**	**13,800 ± 4820**	**4420 ± 2860**	**y = −3350x + 44,000 ***	**0.99**	**Decreasing**
**CL_(72:09)**	**1280 ± 774**	**1150 ± 378**	**798 ± 339**	**456 ± 284**	**y = −105x + 1720 ***	**0.97**	**Decreasing**
CL_(72:10)	66.9 ± 45.2	62.8 ± 21.4	62 ± 40.5	70.8 ± 37.4	y = 0.404x + 62.5	0.12	
**CL_(74:06)**	**1080 ± 745**	**930 ± 448**	**489 ± 203**	**395 ± 366**	**y = −92.4x + 1430 ***	**0.94**	**Decreasing**
**CL_(74:07)**	**4550 ± 2900**	**3350 ± 1190**	**2000 ± 712**	**1650 ± 1280**	**y = −372x + 5740 ***	**0.95**	**Decreasing**
**CL_(74:08)**	**8200 ± 4930**	**6210 ± 1800**	**4510 ± 1460**	**3490 ± 2360**	**y = −586x + 10,100 ***	**0.98**	**Decreasing**
**CL_(74:09)**	**7180 ± 3960**	**6480 ± 1830**	**5090 ± 1530**	**3460 ± 2210**	**y = −465x + 9110 ***	**0.97**	**Decreasing**
**CL_(74:10)**	**3980 ± 2390**	**3050 ± 652**	**1860 ± 595**	**1030 ± 673**	**y = −372x + 5320 ***	**1.00**	**Decreasing**
**CL_(74:11)**	**1240 ± 707**	**784 ± 177**	**454 ± 136**	**284 ± 181**	**y = −118x + 1600 ***	**0.96**	**Decreasing**
CL_(76:09)	802 ± 495	560 ± 202	416 ± 152	633 ± 566	y = −24.1x + 787	0.27	
CL_(76:10)	1300 ± 724	974 ± 335	723 ± 242	960 ± 768	y = −47.1x + 1350	0.48	
CL_(76:11)	1580 ± 894	1140 ± 399	678 ± 217	710 ± 530	y = −114x + 1900	0.87	
**CL_(76:12)**	**1200 ± 704**	**925 ± 337**	**513 ± 163**	**441 ± 328**	**y = −99.6x + 1530 ***	**0.94**	**Decreasing**
DG_(32:0)	194 ± 108	64.5 ± 72.4	75.8 ± 122	99.9 ± 108	y = −10x + 185	0.35	
DG_(34:0)	84.3 ± 167	48.3 ± 145	ND ± ND	ND ± ND	y = -13.3x + 132	1.00	
DG_(34:1)	6950 ± 4570	3290 ± 1420	3300 ± 1600	4690 ± 4060	y = −251x + 6480	0.26	
GM1_(34:0)	1.47 ± 1.39	5.46 ± 4.45	4.11 ± 4.43	3.26 ± 2.36	y = 0.149x + 2.44	0.10	
GM1_(34:1)	4.89 ± 3.76	15.3 ± 11.1	10.6 ± 9.26	9.27 ± 6.24	y = 0.313x + 7.62	0.06	
GM1_(34:1-OH)	0.054 ± 0.0451	0.0988 ± 0.0944	0.0491 ± 0.0531	0.0338 ± 0.0588	y = −0.00409x + 0.0902	0.26	
GM1_(36:0)	ND ± ND	0.0285 ± 0.0603	ND ± ND	ND ± ND			
GM1_(36:1)	ND ± ND	0.157 ± 0.356	0.125 ± 0.158	0.152 ± 0.285	y = −0.000926x + 0.153	0.02	
Hex-Cer_(32:0)	0.578 ± 0.44	0.383 ± 0.558	0.199 ± 0.291	0.332 ± 0.324	y = −0.0341x + 0.634	0.57	
Hex-Cer_(32:1)	ND ± ND	0.0486 ± 0.104	0.0269 ± 0.076	0.0396 ± 0.112	y = −0.00167x + 0.0534	0.17	
Hex-Cer_(34:0-OH)	0.0882 ± 0.264	0.0548 ± 0.164	0.0611 ± 0.173	ND ± ND	y = −0.00502x + 0.0997	0.58	
Hex-Cer_(34:1)	0.565 ± 0.791	1.27 ± 1.2	1.74 ± 1.87	4.68 ± 1.85	y = 0.475x − 1.57	0.84	Increasing
Hex-Cer_(34:1-OH)	0.295 ± 0.719	ND ± ND	0.0372 ± 0.105	0.208 ± 0.587	y = −0.016x + 0.31	0.25	
Hex-Cer_(34:2)	0.32 ± 0.418	0.318 ± 0.598	ND ± ND	0.424 ± 1.2	y = 0.0138x + 0.255	0.88	
**Hex-Cer_(34:2-OH)**	**0.614 ± 0.37**	**0.518 ± 0.375**	**0.39 ± 0.512**	**0.306 ± 0.488**	**y = −0.039x + 0.755 ***	**0.99**	**Decreasing**
Hex-Cer_(35:0)	0.0425 ± 0.127	ND ± ND	0.663 ± 0.389	1.37 ± 1.56	y = 0.157x − 0.579	0.95	
**Hex-Cer_(35:1)**	**0.551 ± 0.625**	**0.649 ± 0.567**	**1.03 ± 0.58**	**1.66 ± 1.57**	**y = 0.137x − 0.0781 ***	**0.91**	**Increasing**
**Hex-Cer_(36:0-OH)**	**2.03 ± 1.06**	**2.58 ± 2.01**	**3.51 ± 1.07**	**4.75 ± 2.01**	**y = 0.337x + 0.642 ***	**0.97**	**Increasing**
Hex-Cer_(36:1)	0.235 ± 0.413	ND ± ND	0.688 ± 0.661	0.264 ± 0.491	y = 0.0151x + 0.274	0.06	
Hex-Cer_(36:2)	ND ± ND	0.0816 ± 0.245	0.289 ± 0.574	0.638 ± 0.942	y = 0.103x − 0.591	0.98	
Hex-Cer_(37:0)	ND ± ND	0.0835 ± 0.167	0.0639 ± 0.181	0.151 ± 0.295	y = 0.0125x − 0.013	0.55	
Hex-Cer_(37:0-OH)	9.56 ± 9.47	14.3 ± 11.8	38 ± 31.6	80.1 ± 67.3	y = 8.72x − 31.2	0.89	Increasing
Hex-Cer_(37:1)	0.0494 ± 0.148	0.167 ± 0.252	0.0497 ± 0.141	ND ± ND	y = 0.0000556x + 0.0884	0.00	
Hex-Cer_(37:2)	2.12 ± 2.98	3.57 ± 5.06	0.717 ± 2.03	0.957 ± 2.02	y = −0.235x + 3.64	0.39	
Hex-Cer_(38:0-OH)	ND ± ND	3.91 ± 5.51	2.8 ± 4.29	7.46 ± 8.35	y = 0.657x − 1.19	0.53	
Hex-Cer_(38:1)	14.8 ± 16	13.4 ± 12.3	22.6 ± 25.1	32.2 ± 25.2	y = 2.27x + 3.35	0.84	
Hex-Cer_(38:2)	0.569 ± 0.899	2.09 ± 1.23	1.81 ± 1.58	0.813 ± 0.906	y = 0.0167x + 1.19	0.01	
Hex-Cer_(39:0-OH)	5.62 ± 3.39	2.48 ± 3.2	4.86 ± 13.8	19.3 ± 21.8	y = 1.61x − 4.24	0.54	
Hex-Cer_(39:2)	20.7 ± 4.71	21.6 ± 7.5	24.1 ± 7.77	32.5 ± 7.01	y = 1.4x + 14	0.83	Increasing
Hex-Cer_(40:0)	10.1 ± 12.2	9.47 ± 10	15.5 ± 15	19.7 ± 9.5	y = 1.29x + 3.82	0.87	
Hex-Cer_(40:0-OH)	ND ± ND	5 ± 15	ND ± ND	ND ± ND			
**Hex-Cer_(40:1)**	**174 ± 204**	**221 ± 120**	**275 ± 249**	**403 ± 206**	**y = 27.4x + 58.3 ***	**0.94**	**Increasing**
Hex-Cer_(40:1-OH)	7.04 ± 2.48	6.38 ± 2.94	7.95 ± 2.57	5.14 ± 1.51	y = −0.153x + 7.8	0.20	
Hex-Cer_(40:2)	0.266 ± 0.178	0.854 ± 0.804	0.806 ± 1.66	0.672 ± 1.21	y = 0.0433x + 0.318	0.32	
Hex-Cer_(40:2-OH)	2.1 ± 1.32	2.17 ± 0.891	2.76 ± 1.9	1.86 ± 1.91	y = −0.00481x + 2.26	0.00	
Hex-Cer_(41:0-OH)	23.9 ± 8.7	33 ± 10.5	26.8 ± 4.88	20.1 ± 7.48	y = −0.652x + 30.9	0.17	
Hex-Cer_(41:1)	4.01 ± 1.31	5.78 ± 1.62	7.62 ± 3.75	17.7 ± 5.68	y = 1.59x − 3.38	0.82	Increasing
Hex-Cer_(41:2)	4.6 ± 1.03	4.78 ± 1.62	3.34 ± 1.5	3.71 ± 1.89	y = −0.152x + 5.27	0.59	
Hex-Cer_(42:0)	0.394 ± 0.271	0.329 ± 0.289	0.466 ± 0.227	0.705 ± 0.432	y = 0.0396x + 0.17	0.71	
Hex-Cer_(42:0-OH)	21.8 ± 14.5	21.3 ± 6.34	30.3 ± 20.7	34.6 ± 14.3	y = 1.76x + 13.6	0.88	
**Hex-Cer_(42:1)**	**14.1 ± 4.31**	**20.5 ± 5.26**	**22.5 ± 10.5**	**34.5 ± 12.3**	**y = 2.34x + 4.99 ***	**0.92**	**Increasing**
**Hex-Cer_(42:2)**	**7.61 ± 2.29**	**11.3 ± 3.91**	**14.3 ± 7.05**	**23.1 ± 7.39**	**y = 1.83x + 0.061 ***	**0.93**	**Increasing**
Hex-Cer_(42:2-OH)	4.99 ± 1.66	5.86 ± 2.35	5.19 ± 1.86	7.37 ± 3.43	y = 0.24x + 4.02	0.60	
Hex-Cer_(43:0)	0.0149 ± 0.0296	0.0155 ± 0.0308	0.145 ± 0.175	0.129 ± 0.153	y = 0.0175x − 0.0576	0.74	
Hex-Cer_(43:0-OH)	0.416 ± 0.255	0.583 ± 0.36	0.338 ± 0.18	0.32 ± 0.277	y = −0.0197x + 0.565	0.33	
Hex-Cer_(43:1)	1.69 ± 0.642	3.13 ± 1.14	3.47 ± 1.46	7.28 ± 2.91	y = 0.634x − 0.955	0.86	Increasing
Hex-Cer_(43:2)	0.866 ± 0.212	0.858 ± 0.321	0.799 ± 0.314	1.24 ± 0.718	y = 0.0394x + 0.64	0.46	
**LPC_(14:0)**	**1.4 ± 0.645**	**1.58 ± 0.732**	**2.09 ± 0.96**	**2.96 ± 1.34**	**y = 0.192x + 0.537 ***	**0.92**	**Increasing**
LPC_(15:0)	6.02 ± 1.49	7.53 ± 2.48	7.69 ± 2.62	8.38 ± 2.62	y = 0.268x + 5.35	0.88	Increasing
**LPC_(16:0)**	**1430 ± 246**	**1560 ± 272**	**1690 ± 300**	**1900 ± 427**	**y = 57x + 1210 ***	**0.98**	**Increasing**
**LPC_(16:1)**	**8.56 ± 4.67**	**11.3 ± 6.3**	**19.6 ± 8.38**	**33.1 ± 10.5**	**y = 3.03x − 5.07 ***	**0.92**	**Increasing**
LPC_(17:0)	64.5 ± 10.8	72.2 ± 13.1	74.9 ± 11.2	75.6 ± 27.5	y = 1.33x + 61.6	0.84	Increasing
LPC_(17:1)	0.395 ± 0.314	0.576 ± 0.537	2.2 ± 1.09	6.08 ± 1.62	y = 0.692x − 2.98	0.83	Increasing
**LPC_(18:0)**	**3720 ± 504**	**3550 ± 634**	**3450 ± 619**	**3300 ± 520**	**y = −50.4x + 3890 ***	**0.99**	**Decreasing**
LPC_(18:1)	252 ± 47.9	247 ± 56.4	332 ± 51.2	549 ± 121	y = 36.1x + 68.5	0.79	
**LPC_(18:2)**	**266 ± 52.6**	**232 ± 46.8**	**217 ± 19.7**	**198 ± 38.2**	**y = −8.11x + 290 ***	**0.97**	**Decreasing**
**LPC_(18:3)**	**80 ± 15.2**	**88.6 ± 22.2**	**93 ± 19.4**	**102 ± 22.5**	**y = 2.61x + 71 ***	**0.98**	**Increasing**
LPC_(18:4)	0.515 ± 0.37	0.758 ± 0.481	1.45 ± 0.744	3.08 ± 1.42	y = 0.311x − 0.926	0.88	Increasing
LPC_(19:0)	47.8 ± 5.54	37.7 ± 12.1	36.2 ± 8.19	40.1 ± 12.9	y = −0.911x + 47.4	0.38	
**LPC_(20:0)**	**45.5 ± 12**	**29 ± 9.84**	**22.8 ± 4.24**	**18.5 ± 5.39**	**y = −3.23x + 53.7 ***	**0.90**	**Decreasing**
LPC_(20:4)	344 ± 121	289 ± 32.6	288 ± 82.8	282 ± 53.7	y = −6.93x + 354	0.69	Decreasing
LPC_(20:5)	26.9 ± 9.49	21.5 ± 4.39	20.2 ± 2.89	21 ± 5.62	y = −0.704x + 27.8	0.65	
LPC_(21:0)	1.11 ± 0.227	0.89 ± 0.305	0.75 ± 0.257	0.896 ± 0.324	y = −0.029x + 1.13	0.46	
LPC_(22:4)	5.12 ± 2.83	3.37 ± 0.794	3.21 ± 0.909	3.52 ± 1	y = −0.184x + 5.21	0.52	
LPC_(22:5)	12.3 ± 9.44	6.01 ± 2.43	5.97 ± 2.51	9.46 ± 4.97	y = −0.317x + 10.9	0.13	
LPC_(22:6)	21.1 ± 13	14.4 ± 3.33	15.4 ± 4.74	26.5 ± 5.75	y = 0.637x + 14.5	0.16	
**LPE_(16:0)**	**9.94 ± 2.69**	**11.4 ± 3.82**	**16.7 ± 4.93**	**25.8 ± 11.7**	**y = 1.96x + 0.977 ***	**0.91**	**Increasing**
LPE_(16:1)	ND ± ND	0.0351 ± 0.0709	0.0116 ± 0.0329	0.177 ± 0.163	y = 0.0263x − 0.162	0.63	
**LPE_(17:0)**	**0.637 ± 0.392**	**0.724 ± 0.401**	**1.1 ± 0.151**	**1.61 ± 0.982**	**y = 0.122x + 0.0842 ***	**0.92**	**Increasing**
LPE_(18:0)	50.7 ± 12.9	50.3 ± 14.6	57.1 ± 12.6	67.7 ± 29.3	y = 2.14x + 40.1	0.84	
LPE_(18:1)	5.94 ± 1.16	4.7 ± 1.21	6.6 ± 1.36	11 ± 3.41	y = 0.633x + 2.22	0.65	
LPE_(18:2)	5.04 ± 3.8	2.76 ± 1.33	2.93 ± 1.15	2.42 ± 1.28	y = −0.285x + 5.47	0.70	
**LPE_(18:3)**	**1.85 ± 0.528**	**2.14 ± 0.76**	**2.96 ± 0.944**	**4.22 ± 1.93**	**y = 0.294x + 0.546 ***	**0.93**	**Increasing**
**LPE_(20:0)**	**0.444 ± 0.114**	**0.358 ± 0.169**	**0.31 ± 0.121**	**0.247 ± 0.17**	**y = −0.0237x + 0.521 ***	**0.99**	**Decreasing**
**LPE_(20:3)**	**0.472 ± 0.645**	**0.556 ± 0.49**	**0.945 ± 0.831**	**1.49 ± 1.22**	**y = 0.128x − 0.11 ***	**0.92**	**Increasing**
LPE_(20:4)	22.9 ± 12.3	17.9 ± 4.53	20.8 ± 7.66	28.6 ± 11.8	y = 0.741x + 16.9	0.33	
LPE_(20:5)	1.53 ± 1.12	0.932 ± 0.388	0.827 ± 0.252	0.645 ± 0.248	y = −0.102x + 1.77	0.86	Decreasing
LPE_(22:4)	0.264 ± 0.791	ND ± ND	0.0209 ± 0.0389	0.35 ± 0.382	y = 0.00267x + 0.19	0.00	
LPI_(16:0)	64.1 ± 21.4	82.6 ± 36.1	113 ± 39.7	103 ± 45	y = 5.45x + 49	0.76	
LPI_(17:0)	5.04 ± 3.17	6.56 ± 3.13	7.63 ± 3.8	5.1 ± 4.85	y = 0.0463x + 5.73	0.02	
LPI_(18:0)	1210 ± 404	1510 ± 647	1510 ± 376	1200 ± 547	y = −1.11x + 1370	0.00	
**LPI_(18:1)**	**46.2 ± 22.3**	**47.6 ± 17.7**	**57.4 ± 21.6**	**59.5 ± 26.8**	**y = 1.84x + 38.6 ***	**0.90**	**Increasing**
LPI_(18:2)	18.9 ± 10.4	17.8 ± 8	16.9 ± 5.86	7.43 ± 4.45	y = −1.31x + 25.3	0.74	Decreasing
LPI_(20:0)	0.0835 ± 0.1	ND ± ND	0.18 ± 0.264	0.0895 ± 0.127	y = 0.00319x + 0.0918	0.06	
**LPI_(20:2)**	**1.7 ± 1.09**	**1.18 ± 1.36**	**1.09 ± 0.93**	**0.661 ± 1.15**	**y = −0.119x + 2.07 ***	**0.94**	**Decreasing**
**LPI_(20:3)**	**25.8 ± 14.7**	**47.1 ± 27.2**	**85.6 ± 35.3**	**125 ± 49.4**	**y = 12.4x − 24.4 ***	**0.98**	**Increasing**
LPI_(20:4)	288 ± 102	453 ± 238	447 ± 108	360 ± 145	y = 7.78x + 328	0.12	
LPI_(22:4)	4.72 ± 2.12	3.96 ± 2.88	4.26 ± 2.1	3.26 ± 1.91	y = −0.151x + 5.21	0.74	
LPI_(22:5)	1.15 ± 1.67	1.51 ± 1.76	0.418 ± 0.701	1.98 ± 1.41	y = 0.0518x + 0.868	0.08	
LPI_(22:6)	2.06 ± 1.55	2.04 ± 1.58	0.605 ± 0.968	2.65 ± 2.37	y = 0.0124x + 1.74	0.00	
Lyso_CL_(52:01)	0.698 ± 1.02	1.75 ± 1.42	1.51 ± 0.755	1.99 ± 1.82	y = 0.135x + 0.457	0.70	
Lyso_CL_(52:02)	11.4 ± 10.4	13.7 ± 7.67	10.7 ± 4.9	12.5 ± 11.6	y = 0.0111x + 12	0.00	
**Lyso_CL_(52:03)**	**50.5 ± 42.8**	**32.9 ± 12.7**	**18.7 ± 8.32**	**17.2 ± 14.1**	**y = −4.23x + 62.2 ***	**0.90**	**Decreasing**
Lyso_CL_(52:04)	30.3 ± 36.6	19.7 ± 9.87	9.23 ± 4.21	10.4 ± 9.01	y = −2.6x + 37.3	0.86	
Lyso_CL_(52:05)	ND ± ND	ND ± ND	ND ± ND	0.785 ± 1.84			
Lyso_CL_(53:04)	ND ± ND	0.154 ± 0.261	0.221 ± 0.404	1.55 ± 1.72	y = 0.259x − 1.69	0.79	
Lyso_CL_(54:02)	ND ± ND	0.0851 ± 0.178	0.0412 ± 0.117	0.655 ± 1.15	y = 0.106x − 0.689	0.69	
Lyso_CL_(54:03)	24.8 ± 26.6	13.2 ± 7.1	9.89 ± 3.88	19.5 ± 18.6	y = −0.711x + 22.3	0.14	
Lyso_CL_(54:04)	172 ± 166	94.9 ± 32.4	67.3 ± 25.2	87 ± 72.4	y = −10.5x + 185	0.63	
Lyso_CL_(54:05)	405 ± 411	234 ± 94	144 ± 60.4	133 ± 104	y = −33.6x + 486	0.87	Decreasing
Lyso_CL_(54:06)	548 ± 555	247 ± 117	122 ± 47.8	65.7 ± 51.8	y = −58.2x + 691	0.89	Decreasing
Lyso_CL_(56:05)	42.9 ± 55.3	13.1 ± 3.27	4.18 ± 2.41	13.5 ± 16.1	y = −3.6x + 45.9	0.55	
Lyso_CL_(56:06)	135 ± 153	42.7 ± 13.4	20.9 ± 8.33	40.2 ± 42.2	y = −11.3x + 146	0.60	
PA_(30:0)	1.19 ± 2.41	2.63 ± 2.77	6.15 ± 4.83	30 ± 23.7	y = 3.33x − 15.5	0.74	Increasing
PA_(30:1)	17.2 ± 17.8	17.1 ± 11.8	20.7 ± 9.98	73.9 ± 41.3	y = 6.43x − 17	0.65	
PA_(32:1)	43.2 ± 46.5	56.3 ± 17.7	84.6 ± 30.3	291 ± 143	y = 28.6x − 99.9	0.74	Increasing
PA_(32:2)	223 ± 216	180 ± 96	154 ± 86.4	166 ± 70.4	y = −7.3x + 237	0.71	
PA_(34:1)	387 ± 401	402 ± 135	538 ± 176	1010 ± 307	y = 74.3x + 16.2	0.79	Increasing
**PA_(34:2)**	**2330 ± 2170**	**1990 ± 756**	**1600 ± 787**	**1230 ± 358**	**y = −137x + 2830 ***	**1.00**	**Decreasing**
PA_(36:1)	1440 ± 924	1640 ± 435	1410 ± 877	1240 ± 323	y = −30.7x + 1670	0.43	
**PA_(36:2)**	**3310 ± 1750**	**2480 ± 918**	**2250 ± 1360**	**1910 ± 1020**	**y = −164x + 3740 ***	**0.92**	**Decreasing**
PA_(36:4)	10,300 ± 9970	6760 ± 2840	7130 ± 3790	9580 ± 4540	y = −66.3x + 8950	0.02	
PA_(38:3)	1390 ± 910	1150 ± 544	1590 ± 683	2590 ± 1230	y = 150x + 535	0.68	
PA_(38:4)	42,400 ± 40,100	27,100 ± 11,700	25,400 ± 17,000	23,000 ± 7630	y = −2220x + 46,400	0.78	Decreasing
PA_(38:5)	3280 ± 2800	2230 ± 911	2430 ± 1180	3310 ± 1300	y = 10.7x + 2730	0.00	
PA_(40:5)	1730 ± 1530	838 ± 336	871 ± 449	1110 ± 412	y = −67.7x + 1650	0.33	
**PC_(30:0)**	**6.27 ± 6.94**	**21.5 ± 23.9**	**39.3 ± 18.4**	**81.4 ± 23.1**	**y = 9.01x − 31.8 ***	**0.94**	**Increasing**
PC_(30:1)	8.63 ± 6.24	12.9 ± 8.52	40.6 ± 32.6	188 ± 120	y = 21x − 97.8	0.74	Increasing
PC_(31:0)	54.7 ± 17.4	78.7 ± 33.5	85.4 ± 22.6	179 ± 44.5	y = 14.1x − 8.1	0.80	Increasing
PC_(32:0)	1750 ± 251	1930 ± 493	1650 ± 237	1950 ± 234	y = 11.9x + 1730	0.08	
PC_(32:1)	222 ± 110	352 ± 101	1240 ± 835	4860 ± 1550	y = 548x − 2530	0.77	Increasing
PC_(32:2)	565 ± 249	496 ± 222	646 ± 223	1350 ± 659	y = 92.8x + 54.5	0.67	
PC_(33:0)	28.2 ± 8.55	39.9 ± 13	49.6 ± 7.5	102 ± 25.7	y = 8.56x − 10.6	0.84	Increasing
PC_(33:1)	148 ± 45.4	267 ± 71.4	540 ± 136	1700 ± 388	y = 183x − 733	0.80	Increasing
PC_(33:2)	470 ± 56.2	515 ± 187	574 ± 241	566 ± 180	y = 12.9x + 433	0.85	
PC_(34:0)	789 ± 267	732 ± 165	615 ± 45.6	644 ± 83.3	y = −20.4x + 851	0.79	
PC_(34:1)	4160 ± 763	5580 ± 931	8080 ± 829	15,100 ± 1050	y = 1310x − 1780	0.88	Increasing
PC_(34:2)	11,900 ± 2050	14,700 ± 1880	15,200 ± 1930	20,400 ± 2030	y = 963x + 8180	0.90	Increasing
PC_(34:3)	3270 ± 1090	3620 ± 834	5400 ± 998	9050 ± 1860	y = 708x − 82.3	0.87	Increasing
PC_(35:0)	28.3 ± 9.27	32.3 ± 9.11	38.3 ± 6.8	71.2 ± 15.2	y = 4.99x + 4.36	0.79	Increasing
PC_(35:1)	108 ± 32	166 ± 36.7	297 ± 52.6	891 ± 153	y = 91.9x − 337	0.79	Increasing
PC_(35:2)	1240 ± 238	1530 ± 305	1530 ± 433	1640 ± 369	y = 44.4x + 1150	0.82	
PC_(36:0)	76.2 ± 27.6	66.4 ± 16.6	55.2 ± 22.9	83.4 ± 16.5	y = 0.385x + 67.4	0.01	
PC_(36:1)	3060 ± 753	3610 ± 756	5450 ± 944	11,600 ± 974	y = 1020x − 1850	0.82	Increasing
PC_(36:2)	6650 ± 1180	7290 ± 1320	7890 ± 1490	10,700 ± 954	y = 472x + 4520	0.85	Increasing
PC_(36:3)	8480 ± 912	9060 ± 1570	11,800 ± 1920	23,900 ± 2150	y = 1810x − 573	0.77	Increasing
PC_(36:4)	18,900 ± 3120	20,200 ± 2700	19,500 ± 1190	27,500 ± 1900	y = 930x + 14,400	0.65	
PC_(37:0)	0.557 ± 0.4	0.542 ± 0.232	0.338 ± 0.231	0.689 ± 0.551	y = 0.00711x + 0.477	0.03	
PC_(37:1)	40.9 ± 20.8	38 ± 19.4	75.1 ± 38.1	283 ± 96.2	y = 28.3x − 107	0.71	
PC_(37:2)	301 ± 88.2	217 ± 62.9	210 ± 61.6	301 ± 56.6	y = −0.259x + 259	0.00	
**PC_(37:3)**	**83.4 ± 17.3**	**86.2 ± 25.1**	**131 ± 33.9**	**160 ± 31.9**	**y = 10.2x + 37.3 ***	**0.92**	**Increasing**
PC_(37:4)	2810 ± 648	3180 ± 738	2520 ± 674	2430 ± 461	y = −66.7x + 3250	0.47	
PC_(37:5)	157 ± 59.9	171 ± 36.4	270 ± 103	445 ± 77.3	y = 35.7x − 12.1	0.88	Increasing
PC_(37:6)	85 ± 34.3	76 ± 37.5	55.8 ± 29.9	101 ± 29.7	y = 1.03x + 71.6	0.04	
PC_(38:0)	1.83 ± 0.323	1.77 ± 0.432	1.79 ± 0.773	2.67 ± 0.91	y = 0.0941x + 1.3	0.56	
PC_(38:1)	39.4 ± 20.9	14.1 ± 12.8	28.9 ± 21.3	103 ± 42.3	y = 7.61x − 11.9	0.46	
PC_(38:2)	649 ± 137	521 ± 186	493 ± 137	871 ± 171	y = 23.6x + 453	0.23	
PC_(38:3)	3030 ± 346	3260 ± 708	5020 ± 1050	12,200 ± 1770	y = 1080x − 2420	0.77	Increasing
PC_(38:4)	10,300 ± 2010	11,700 ± 2240	10,200 ± 1090	12,100 ± 2290	y = 144x + 9970	0.27	
PC_(38:5)	6010 ± 2050	5500 ± 1900	5290 ± 916	8730 ± 1930	y = 294x + 4130	0.41	
PC_(38:6)	16,000 ± 3530	14,000 ± 1550	12,900 ± 1690	18,900 ± 2600	y = 281x + 13,300	0.14	
PC_(40:2)	19.1 ± 8.48	14.8 ± 4.37	13.2 ± 3.21	20.1 ± 5.31	y = 0.0519x + 16.4	0.00	
PC_(40:3)	52.8 ± 15	44 ± 15.1	52.7 ± 13.6	124 ± 22.6	y = 8.23x + 5.39	0.59	
**PC_(40:4)**	**1200 ± 402**	**884 ± 230**	**726 ± 182**	**665 ± 175**	**y = −65.3x + 1370 ***	**0.90**	**Decreasing**
PC_(40:5)	947 ± 577	704 ± 574	514 ± 208	980 ± 459	y = −3.37x + 812	0.00	
PC_(40:6)	3800 ± 936	3630 ± 913	3090 ± 720	4370 ± 960	y = 43.3x + 3390	0.08	
PC_C18(plas)-18:1	ND ± ND	ND ± ND	ND ± ND	0.766 ± 1.43			
PE_(30:0)	4.78 ± 10.7	2.19 ± 4.47	ND ± ND	1.82 ± 5.14	y = −0.323x + 5.26	0.68	
PE_(32:1)	12.4 ± 20.9	26 ± 38.5	468 ± 553	1920 ± 896	y = 228x − 1140	0.78	Increasing
PE_(34:0)	80.3 ± 34.4	106 ± 24	54.8 ± 20.8	30.5 ± 35.6	y = −7.43x + 125	0.63	
PE_(34:1)	4110 ± 2220	4930 ± 1800	9030 ± 1940	18,100 ± 6000	y = 1710x − 4010	0.86	Increasing
**PE_(34:2)**	**71,600 ± 27,000**	**69,300 ± 42,400**	**48,600 ± 14,200**	**29,500 ± 12,000**	**y = −5440x + 96,400 ***	**0.92**	**Decreasing**
**PE_(35:1)**	**4.9 ± 2**	**10.6 ± 3.84**	**23.9 ± 7.89**	**42.4 ± 13**	**y = 4.66x − 15.2 ***	**0.95**	**Increasing**
PE_(36:0)	24.4 ± 11.3	35 ± 15	22.8 ± 6.68	17 ± 12.8	y = −1.27x + 34.5	0.35	
PE_(36:1)	6560 ± 4060	5730 ± 2600	6590 ± 1730	11,300 ± 4870	y = 559x + 3270	0.59	
**PE_(36:2)**	**46,300 ± 18,900**	**36,900 ± 20,900**	**24,400 ± 6230**	**19,300 ± 7930**	**y = −3460x + 58,200 ***	**0.98**	**Decreasing**
**PE_(36:3)**	**39,000 ± 15,000**	**28,800 ± 17,100**	**20,000 ± 4520**	**14,300 ± 5490**	**y = −3070x + 49,000 ***	**0.99**	**Decreasing**
PE_(36:4)	110,000 ± 44,700	107,000 ± 49,200	96,600 ± 12,500	126,000 ± 34,500	y = 1390x + 99,200	0.16	
PE_(38:1)	219 ± 130	144 ± 84.5	115 ± 22	121 ± 46.7	y = −12x + 241	0.76	
PE_(38:2)	1530 ± 645	822 ± 467	539 ± 98.4	535 ± 140	y = −121x + 1780	0.81	Decreasing
PE_(38:3)	13400 ± 4930	13,000 ± 7700	14,400 ± 2710	17,700 ± 3860	y = 530x + 10,600	0.75	
PE_(38:4)	165,000 ± 74,800	148,000 ± 59,300	113,000 ± 20,200	118,000 ± 26,000	y = −6520x + 186,000	0.84	
PE_(38:5)	72,900 ± 38,400	52,700 ± 23,200	39,700 ± 7150	44,700 ± 11,600	y = −3610x + 80,200	0.74	
PE_(38:6)	25,000 ± 12,000	18,200 ± 8270	12,300 ± 2890	19,100 ± 5400	y = −874x + 25,300	0.34	
PG_(32:0)	1.64 ± 0.748	1.59 ± 0.834	5.29 ± 4.05	11 ± 3.46	y = 1.18x − 4.12	0.86	
PG_(33:0)	ND ± ND	ND ± ND	ND ± ND	0.0213 ± 0.0434			
PG_(34:1)	ND ± ND	ND ± ND	2.24 ± 3.19	65.1 ± 51.8	y = 23.3x − 207	1.00	
**PG_(35:1)**	**0.256 ± 0.325**	**0.467 ± 0.67**	**0.816 ± 0.888**	**1.62 ± 0.913**	**y = 0.164x − 0.469 ***	**0.91**	**Increasing**
PG_(36:0)	11.8 ± 3.75	25.6 ± 18	13.4 ± 4.49	7.6 ± 5.1	y = −0.919x + 21.6	0.17	
PG_(36:1)	0.0408 ± 0.122	1.84 ± 2.72	0.525 ± 0.62	0.294 ± 0.831	y = −0.0206x + 0.832	0.01	
PG_(36:2)	125 ± 25.3	169 ± 81.3	106 ± 43.1	53.5 ± 21	y = −10.3x + 192	0.56	
PG_(36:3)	48.8 ± 14.2	62.3 ± 30	32.3 ± 9.53	28.1 ± 20.5	y = −3.41x + 69	0.57	
PG_(36:4)	25.6 ± 13.6	45.9 ± 26.9	33.6 ± 15.7	29.1 ± 15.4	y = −0.0667x + 34.1	0.00	
PG_(38:3)	1.72 ± 1.42	1.04 ± 0.753	3.19 ± 3.43	7.34 ± 3.64	y = 0.704x − 2.06	0.75	
PG_(38:4)	11.7 ± 5.76	19.5 ± 9.9	13.3 ± 5.22	11.9 ± 5.31	y = −0.207x + 15.7	0.04	
PG_(38:5)	6.31 ± 2.82	9.55 ± 5.47	6.6 ± 2.34	7.16 ± 4.03	y = −0.0148x + 7.52	0.00	
PG_(38:6)	0.0292 ± 0.0876	0.0656 ± 0.0754	0.0529 ± 0.067	0.0457 ± 0.0527	y = 0.00136x + 0.0379	0.10	
PG_(40:6)	0.306 ± 0.378	0.211 ± 0.434	0.028 ± 0.0792	0.773 ± 0.913	y = 0.0451x − 0.0156	0.25	
PG_(42:07)	0.469 ± 0.411	0.424 ± 0.714	0.269 ± 0.332	0.305 ± 0.316	y = −0.024x + 0.55	0.77	
PG_(42:08)	0.0147 ± 0.0192	0.0893 ± 0.11	0.112 ± 0.116	ND ± ND	y = 0.018x − 0.0415	0.91	Increasing
PG_(42:10)	0.0638 ± 0.191	0.314 ± 0.943	ND ± ND	0.0417 ± 0.118	y = -0.00954x + 0.209	0.07	
PI_(34:0)	455 ± 385	482 ± 396	354 ± 237	318 ± 196	y = −20x + 555	0.78	
PI_(34:1)	5070 ± 1420	5070 ± 909	7620 ± 1620	10,100 ± 2670	y = 653x + 1970	0.89	
PI_(34:2)	6510 ± 3850	9740 ± 4140	9450 ± 3880	3260 ± 1640	y = −372x + 10,100	0.18	
PI_(35:0)	0.0814 ± 0.244	0.0917 ± 0.275	ND ± ND	ND ± ND	y = 0.00381x + 0.0677	1.00	
PI_(35:2)	134 ± 112	319 ± 164	261 ± 137	82.5 ± 68.9	y = −7.87x + 259	0.06	
**PI_(36:1)**	**186 ± 165**	**450 ± 292**	**659 ± 339**	**814 ± 589**	**y = 77.5x − 65.8 ***	**0.99**	**Increasing**
PI_(36:2)	3680 ± 2670	5400 ± 2720	4920 ± 2300	2020 ± 1140	y = −202x + 5550	0.22	
PI_(36:4)	16,000 ± 6720	25,800 ± 10,600	27,900 ± 8740	21,000 ± 9500	y = 633x + 17,800	0.17	
PI_(38:3)	6340 ± 4020	10,500 ± 6710	14,200 ± 8060	6420 ± 3950	y = 146x + 8250	0.02	
PI_(38:4)	86,600 ± 41,000	124,000 ± 50,900	119,000 ± 34,000	79,600 ± 40,400	y = −963x + 110,000	0.02	
PI_(38:5)	11,100 ± 5400	15,000 ± 5750	16,600 ± 4890	12,400 ± 5840	y = 204x + 12,200	0.08	
**PI_(40:3)**	**120 ± 56.6**	**94 ± 51.8**	**49.8 ± 19**	**25.2 ± 8.08**	**y = −12.2x + 165 ***	**0.99**	**Decreasing**
PI_(40:4)	1040 ± 615	1830 ± 1050	1380 ± 608	885 ± 607	y = −33.9x + 1540	0.08	
PI_(40:6)	4080 ± 1020	5270 ± 1350	4820 ± 901	6870 ± 1270	y = 293x + 3020	0.75	
PI_(40:8)	42.9 ± 25	48.9 ± 23.4	47.4 ± 22.1	50.8 ± 21	y = 0.822x + 41.2	0.72	
PS_(32:0)	0.673 ± 0.587	0.661 ± 0.918	0.321 ± 0.135	0.802 ± 0.336	y = 0.00174x + 0.601	0.00	
PS_(32:1)	0.00219 ± 0.00658	0.0339 ± 0.0329	0.00448 ± 0.0127	0.00374 ± 0.0106	y = −0.000917x + 0.0181	0.04	
PS_(33:1)	1.25 ± 0.504	1.76 ± 0.541	1.8 ± 0.553	1.19 ± 1.06	y = −0.00519x + 1.54	0.00	
PS_(34:0)	0.0337 ± 0.101	0.291 ± 0.452	0.16 ± 0.211	0.0675 ± 0.096	y = −0.0011x + 0.146	0.00	
PS_(34:1)	ND ± ND	0.0315 ± 0.0945	ND ± ND	0.73 ± 0.911	y = 0.129x − 0.783	1.00	
PS_(34:2)	10.3 ± 4.05	13.8 ± 4.69	15.3 ± 4.34	11.7 ± 5.1	y = 0.211x + 11.2	0.11	
PS_(34:3)	0.051 ± 0.0821	0.0582 ± 0.115	0.202 ± 0.314	0.911 ± 0.888	y = 0.101x − 0.466	0.74	Increasing
PS_(35:2)	0.215 ± 0.514	0.379 ± 0.946	0.0732 ± 0.207	0.101 ± 0.286	y = −0.024x + 0.376	0.36	
PS_(36:0)	0.242 ± 0.479	0.34 ± 0.527	0.261 ± 0.489	0.485 ± 0.672	y = 0.0241x + 0.148	0.58	
PS_(36:1)	63.9 ± 28.7	135 ± 76.6	101 ± 23.5	75.4 ± 43	y = 0.0185x + 93.7	0.00	
PS_(36:2)	115 ± 47	136 ± 51.5	123 ± 33.2	83 ± 23.2	y = −4.04x + 145	0.39	
PS_(38:2)	0.434 ± 0.578	1.66 ± 1.78	0.712 ± 0.42	0.536 ± 0.84	y = −0.0238x + 1.02	0.02	
PS_(38:5)	ND ± ND	0.0745 ± 0.224	ND ± ND	ND ± ND			
PS_(38:6)	113 ± 23.8	126 ± 29.7	140 ± 34.3	201 ± 55.3	y = 10.3x + 66.2	0.85	Increasing
PS_(40:0)	0.352 ± 0.406	1.18 ± 0.599	0.755 ± 0.566	1.46 ± 0.838	y = 0.107x + 0.115	0.59	
PS_(40:2)	0.485 ± 1.31	1.23 ± 2.17	ND ± ND	ND ± ND	y = 0.276x − 0.508	1.00	
PS_(40:3)	215 ± 36.3	243 ± 49.1	206 ± 51.9	147 ± 28.2	y = −8.93x + 271	0.59	
PS_(42:6)	185 ± 75.9	262 ± 104	237 ± 95.8	87.8 ± 37.6	y = −11.7x + 283	0.28	
S_(32:0)	0.134 ± 0.0379	0.128 ± 0.0538	0.112 ± 0.0157	0.0075 ± 0.014	y = −0.0146x + 0.207	0.74	Decreasing
S_(32:1)	ND ± ND	ND ± ND	ND ± ND	0.000566 ± 0.0016			
S_(34:0)	1.04 ± 0.105	2.09 ± 1.77	1.54 ± 0.259	1.31 ± 0.586	y = 0.00963x + 1.42	0.01	
S_(34:1-OH)	1.47 ± 0.48	1.72 ± 0.795	1.08 ± 0.388	1.35 ± 0.9	y = −0.037x + 1.69	0.24	
S_(34:2)	0.592 ± 0.147	0.769 ± 0.36	0.402 ± 0.187	0.327 ± 0.128	y = −0.043x + 0.852	0.57	
S_(35:0)	0.0316 ± 0.0617	0.022 ± 0.0369	0.0626 ± 0.123	0.0338 ± 0.0371	y = 0.00175x + 0.0241	0.12	
S_(35:1)	0.0235 ± 0.041	0.021 ± 0.0344	0.0404 ± 0.0922	0.16 ± 0.307	y = 0.0159x − 0.0603	0.70	
S_(35:1-OH)	0.209 ± 0.281	0.127 ± 0.381	ND ± ND	ND ± ND	y = −0.0304x + 0.318	1.00	
S_(35:2)	ND ± ND	ND ± ND	0.00482 ± 0.00897	0.00385 ± 0.00881	y = −0.000359x + 0.00805	1.00	
S_(36:1)	0.999 ± 0.437	1 ± 0.661	0.824 ± 0.375	0.561 ± 0.391	y = −0.0552x + 1.27	0.86	
S_(37:2)	0.159 ± 0.092	0.296 ± 0.22	0.206 ± 0.125	0.08 ± 0.0753	y = −0.0121x + 0.278	0.22	
S_(38:0)	1.88 ± 0.631	3.39 ± 2.13	1.89 ± 0.684	1.03 ± 0.594	y = −0.15x + 3.2	0.28	
S_(38:1-OH)	0.153 ± 0.242	0.197 ± 0.308	0.116 ± 0.253	0.0765 ± 0.109	y = −0.0115x + 0.224	0.61	
S_(39:1)	0.111 ± 0.0948	0.41 ± 0.632	0.194 ± 0.249	0.0621 ± 0.12	y = −0.0134x + 0.297	0.09	
S_(40:0)	7.77 ± 1.43	17.9 ± 12.7	10.5 ± 4.51	7.67 ± 4.6	y = −0.285x + 13.1	0.04	
S_(40:1-OH)	1.48 ± 0.815	1.78 ± 1.37	1.57 ± 0.909	1.46 ± 1.39	y = −0.01x + 1.65	0.06	
S_(40:2)	1.04 ± 0.637	1.44 ± 0.841	1.03 ± 0.425	0.566 ± 0.349	y = −0.0679x + 1.54	0.44	
S_(41:0)	0.352 ± 0.195	1.1 ± 0.943	0.253 ± 0.162	0.0222 ± 0.0413	y = −0.068x + 0.952	0.26	
S_(41:2)	ND ± ND	0.0108 ± 0.0324	0.021 ± 0.0318	0.0543 ± 0.0655	y = 0.00806x − 0.0438	0.91	
S_(42:0)	0.322 ± 0.225	0.59 ± 0.876	0.491 ± 0.223	0.294 ± 0.162	y = −0.00678x + 0.476	0.03	
S_(42:2)	0.395 ± 0.385	1.1 ± 1.26	0.958 ± 0.837	0.983 ± 0.763	y = 0.0601x + 0.399	0.44	
S_(42:2-OH)	2.43 ± 0.843	1.55 ± 0.588	0.786 ± 0.406	0.789 ± 0.434	y = −0.211x + 3	0.88	
S_(46:2-OH)	0.0599 ± 0.0337	0.0149 ± 0.0269	0.0228 ± 0.018	0.0258 ± 0.031	y = −0.0035x + 0.0576	0.37	
S_(48:2-OH)	0.00958 ± 0.0163	ND ± ND	0.00395 ± 0.00731	0.00115 ± 0.00326	y = −0.00104x + 0.0133	1.00	
SM_(30:1)	1.61 ± 0.262	1.59 ± 0.611	1.89 ± 0.679	1.84 ± 0.667	y = 0.0367x + 1.45	0.68	
**SM_(32:0)**	**4.99 ± 0.941**	**4.66 ± 2.13**	**3.81 ± 1.08**	**2.85 ± 1.21**	**y = −0.269x + 6.14 ***	**0.96**	**Decreasing**
**SM_(32:1)**	**428 ± 88.3**	**368 ± 107**	**334 ± 87.3**	**256 ± 63.6**	**y = −20.4x + 502 ***	**0.98**	**Decreasing**
**SM_(33:1)**	**273 ± 37.3**	**238 ± 62.2**	**168 ± 32.2**	**135 ± 44.4**	**y = −17.9x + 341 ***	**0.98**	**Decreasing**
SM_(34:0)	21.8 ± 5.98	26 ± 6.49	22.2 ± 6.65	18 ± 4.85	y = −0.563x + 26.3	0.36	
**SM_(34:0-OH)**	**7.39 ± 2.3**	**5.26 ± 2.46**	**3.5 ± 1.7**	**1.4 ± 1.69**	**y = −0.731x + 9.98 ***	**1.00**	**Decreasing**
SM_(34:1)	3400 ± 495	3830 ± 796	3240 ± 427	2620 ± 346	y = −109x + 4100	0.57	
SM_(34:1-OH)	4.36 ± 1.63	10.1 ± 8	5.89 ± 3.22	2.74 ± 2.48	y = −0.336x + 8.34	0.14	
SM_(34:2)	397 ± 102	401 ± 106	320 ± 50.4	282 ± 94.6	y = −15.8x + 471	0.88	
**SM_(34:2-OH)**	**4.44 ± 1.37**	**4.15 ± 1.73**	**3.2 ± 0.88**	**2.68 ± 0.928**	**y = −0.231x + 5.38 ***	**0.96**	**Decreasing**
SM_(35:0)	0.957 ± 1.35	0.969 ± 0.848	0.457 ± 0.267	0.389 ± 0.404	y = −0.0821x + 1.32	0.84	
SM_(35:1)	36.3 ± 3.74	48.6 ± 6.53	40.9 ± 7.5	41.1 ± 8.38	y = 0.248x + 39.8	0.03	
SM_(35:2)	0.632 ± 0.456	0.335 ± 0.314	0.144 ± 0.149	0.137 ± 0.18	y = −0.0621x + 0.787	0.87	Decreasing
SM_(36:1)	449 ± 43.4	529 ± 103	446 ± 58.3	298 ± 50.5	y = −19.9x + 582	0.52	
SM_(36:2)	32 ± 6.24	38.6 ± 9.35	30.9 ± 4.52	23.1 ± 5.49	y = −1.27x + 40.9	0.49	
**SM_(36:3)**	**2.19 ± 0.784**	**1.71 ± 0.891**	**0.76 ± 0.384**	**0.406 ± 0.326**	**y = −0.233x + 3.05 ***	**0.97**	**Decreasing**
SM_(37:2)	0.0328 ± 0.0984	0.024 ± 0.0721	ND ± ND	0.0197 ± 0.0556	y = −0.0015x + 0.0363	0.86	
SM_(38:0)	7.73 ± 1.45	7.83 ± 3.24	6.35 ± 1.81	4.64 ± 1.6	y = −0.398x + 9.68	0.86	
**SM_(38:1)**	**342 ± 56.4**	**293 ± 59.3**	**200 ± 27.7**	**106 ± 19.4**	**y = −29.7x + 462 ***	**0.98**	**Decreasing**
SM_(39:0)	8.55 ± 3.43	8.71 ± 6.78	12.5 ± 7.79	ND ± ND	y = 0.731x + 5.31	0.78	Increasing
**SM_(39:1)**	**178 ± 28.6**	**142 ± 35.5**	**109 ± 28.1**	**85.4 ± 10.7**	**y = −11.5x + 217 ***	**0.99**	**Decreasing**
**SM_(40:0)**	**17.3 ± 5.88**	**14.2 ± 4.27**	**10.8 ± 2.35**	**4.03 ± 1.2**	**y = −1.6x + 23.8 ***	**0.96**	**Decreasing**
**SM_(40:0-OH)**	**5.96 ± 1.62**	**4.29 ± 1.17**	**3.53 ± 1.12**	**0.919 ± 0.868**	**y = −0.588x + 8.17 ***	**0.95**	**Decreasing**
**SM_(40:1)**	**989 ± 137**	**867 ± 181**	**704 ± 125**	**520 ± 71.7**	**y = −58.1x + 1210 ***	**0.99**	**Decreasing**
**SM_(40:2)**	**138 ± 21.3**	**123 ± 21.6**	**91.7 ± 16.9**	**55.3 ± 11.1**	**y = −10.3x + 181 ***	**0.97**	**Decreasing**
SM_(41:0)	8.36 ± 2.45	8.68 ± 2.13	7.03 ± 1.56	5.33 ± 1.63	y = −0.398x + 10.4	0.83	
**SM_(41:1)**	**893 ± 123**	**839 ± 144**	**772 ± 135**	**695 ± 133**	**y = −24.5x + 987 ***	**0.99**	**Decreasing**
SM_(42:0)	8.67 ± 2.47	10.5 ± 3.94	7.13 ± 2.87	3.17 ± 1.49	y = −0.736x + 13	0.68	
**SM_(42:0-OH)**	**12.7 ± 2.47**	**11.1 ± 1.98**	**9.01 ± 2.88**	**4.92 ± 1.53**	**y = −0.942x + 16.6 ***	**0.95**	**Decreasing**
**SM_(42:1)**	**1610 ± 185**	**1540 ± 227**	**1280 ± 206**	**1020 ± 194**	**y = −75.2x + 1940 ***	**0.95**	**Decreasing**
SM_(42:2)	1160 ± 180	1180 ± 185	1010 ± 197	880 ± 109	y = −37.4x + 1340	0.86	
SM_(43:0)	0.202 ± 0.119	0.527 ± 0.275	0.368 ± 0.199	0.115 ± 0.152	y = −0.0156x + 0.422	0.09	
SM_(43:1)	158 ± 43.1	200 ± 35.8	163 ± 44.2	153 ± 57	y = −1.93x + 183	0.10	
SM_(44:1)	13.5 ± 4.24	17.9 ± 3.6	14.1 ± 4.99	9.32 ± 3.95	y = −0.605x + 18.3	0.36	
SM_(44:2)	27 ± 6.17	29.6 ± 5.31	27 ± 7.84	23.1 ± 5.83	y = −0.53x + 30.7	0.47	
TG_(18:0)	11 ± 8.91	11.3 ± 4.51	6.54 ± 4.04	2.47 ± 2.6	y = −1.12x + 16.4	0.88	
TG_(24:0)	920 ± 2500	39.9 ± 59.6	84.9 ± 195	177 ± 372	y = −80.9x + 924	0.46	
TG_(36:0)	170 ± 214	52.4 ± 36.9	93.4 ± 83.8	441 ± 288	y = 31.6x − 52.8	0.40	
TG_(44:1)	970 ± 941	325 ± 175	1550 ± 2130	2250 ± 1250	y = 188x − 161	0.63	
TG_(45:1)	14.3 ± 19	7.72 ± 5.87	32.3 ± 32.3	83.7 ± 49.3	y = 8.62x − 31.4	0.76	
TG_(45:2)	92.6 ± 86.2	31.6 ± 14.8	64.1 ± 58.7	84.2 ± 53.8	y = 0.27x + 66.1	0.00	
TG_(46:1)	85.4 ± 104	22.5 ± 16.1	87.6 ± 112	269 ± 189	y = 22.8x − 58.4	0.56	
TG_(46:2)	1060 ± 1040	293 ± 167	412 ± 311	787 ± 609	y = -25.9x + 836	0.07	
TG_(46:4)	3530 ± 3550	700 ± 350	464 ± 357	316 ± 241	y = −366x + 4050	0.70	Decreasing
TG_(47:0)	7.11 ± 7	2.59 ± 2.23	2.86 ± 4.1	6.91 ± 2.88	y = −0.0122x + 4.96	0.00	
TG_(47:1)	1.39 ± 3.19	0.253 ± 0.76	6.65 ± 10.2	19.6 ± 9.01	y = 2.26x − 10.3	0.79	
TG_(47:2)	11.5 ± 11.3	3.39 ± 2.87	10.9 ± 6.94	22.3 ± 17.7	y = 1.48x + 0.715	0.44	
TG_(48:0)	133 ± 69.6	63 ± 10.9	75.7 ± 98.4	92.4 ± 43.9	y = −4.04x + 122	0.21	
TG_(48:1)	142 ± 100	83 ± 29.2	279 ± 343	435 ± 172	y = 39.8x − 69.8	0.78	
TG_(48:2)	241 ± 196	115 ± 38.3	319 ± 401	440 ± 165	y = 29.7x + 51.8	0.57	
TG_(48:3)	250 ± 185	83.7 ± 27	112 ± 80.1	102 ± 49.9	y = −15.4x + 255	0.49	
TG_(49:0)	18 ± 9.16	10.7 ± 2.12	11 ± 5.69	10.2 ± 3.69	y = −0.856x + 19	0.65	
TG_(49:1)	35.7 ± 18.6	28.3 ± 9.67	67.9 ± 55.5	96.3 ± 30.3	y = 8.2x − 5.68	0.83	
TG_(49:2)	70.2 ± 29.8	50.1 ± 12.1	77.6 ± 55.4	86.3 ± 30.3	y = 2.81x + 49.6	0.40	
TG_(49:3)	27.3 ± 12.9	18.2 ± 5.03	26.8 ± 15.4	19.3 ± 7.36	y = −0.57x + 27.3	0.17	
TG_(50:0)	178 ± 68.6	118 ± 20.3	125 ± 57.6	102 ± 37.6	y = −8.19x + 193	0.75	
TG_(50:1)	1440 ± 548	1030 ± 339	1870 ± 1370	1870 ± 858	y = 78.9x + 949	0.47	
TG_(50:2)	2750 ± 1210	1830 ± 458	2920 ± 2200	2920 ± 1010	y = 59.3x + 2150	0.16	
TG_(50:3)	1500 ± 858	1050 ± 283	1760 ± 1370	1040 ± 452	y = −24.8x + 1530	0.06	
TG_(51:1)	79 ± 30.9	78.4 ± 31.8	138 ± 102	124 ± 39.6	y = 7.21x + 49.7	0.67	
TG_(51:2)	296 ± 163	274 ± 82.5	532 ± 293	509 ± 148	y = 33.2x + 149	0.72	
**TG_(51:3)**	**648 ± 459**	**467 ± 133**	**437 ± 175**	**186 ± 86.3**	**y = −52.4x + 836 ***	**0.92**	**Decreasing**
**TG_(51:4)**	**662 ± 568**	**330 ± 121**	**150 ± 55.5**	**37.2 ± 17.5**	**y = −76.1x + 877 ***	**0.94**	**Decreasing**
**TG_(52:0)**	**128 ± 70.4**	**80.4 ± 17.6**	**69.5 ± 28.8**	**47.4 ± 24.1**	**y = −9.36x + 153 ***	**0.92**	**Decreasing**
TG_(52:1)	849 ± 304	716 ± 311	1130 ± 904	823 ± 275	y = 12.4x + 784	0.06	
TG_(52:2)	12,300 ± 6460	9670 ± 3230	12,700 ± 6000	8610 ± 3300	y = −298x + 13,100	0.27	
**TG_(52:3)**	**36,800 ± 25,800**	**22,700 ± 5870**	**15,800 ± 5770**	**5160 ± 2670**	**y = −3770x + 49,000 ***	**0.98**	**Decreasing**
**TG_(52:4)**	**45,200 ± 36,700**	**21,100 ± 7240**	**9760 ± 3330**	**2220 ± 1880**	**y = −5200x + 59,300 ***	**0.93**	**Decreasing**
TG_(53:1)	28.5 ± 8.42	23.8 ± 10.3	36.4 ± 29.3	25.5 ± 8.57	y = 0.133x + 27.5	0.01	
TG_(53:2)	253 ± 129	240 ± 109	353 ± 259	214 ± 67.8	y = −0.148x + 266	0.00	
**TG_(53:3)**	**801 ± 527**	**518 ± 145**	**470 ± 175**	**183 ± 73.3**	**y = −70.4x + 1030 ***	**0.94**	**Decreasing**
**TG_(53:4)**	**877 ± 722**	**489 ± 180**	**277 ± 100**	**75.2 ± 38.5**	**y = −96.9x + 1170 ***	**0.97**	**Decreasing**
TG_(54:0)	20.1 ± 12.8	11.5 ± 2.27	9.51 ± 3.73	7.25 ± 5.07	y = −1.5x + 23.6	0.87	Decreasing
TG_(54:1)	117 ± 54.8	79 ± 24.1	96.6 ± 86.6	57.2 ± 25.8	y = −5.99x + 133	0.67	
TG_(54:2)	866 ± 326	637 ± 373	805 ± 731	408 ± 146	y = −44.7x + 1020	0.58	
**TG_(54:3)**	**5400 ± 3270**	**2880 ± 1140**	**2350 ± 1060**	**1000 ± 436**	**y = −509x + 6800 ***	**0.93**	**Decreasing**
**TG_(54:4)**	**14,200 ± 10,600**	**6230 ± 2180**	**3450 ± 1230**	**977 ± 444**	**y = −1570x + 18,200 ***	**0.91**	**Decreasing**
TG_(54:5)	26,100 ± 21,100	10,200 ± 4050	4300 ± 1350	1010 ± 595	y = −3010x + 33,400	0.89	Decreasing
TG_(54:6)	24,900 ± 20,300	9310 ± 3280	2980 ± 1090	554 ± 426	y = −2940x + 31,900	0.88	Decreasing
TG_(55:2)	13.8 ± 4.81	9.88 ± 6.38	16.2 ± 17.2	8.35 ± 3.23	y = −0.371x + 14.9	0.13	
**TG_(55:3)**	**77.5 ± 39.7**	**46.2 ± 18.2**	**44.1 ± 29.4**	**19.7 ± 8.29**	**y = −6.5x + 96.6 ***	**0.91**	**Decreasing**
**TG_(55:4)**	**142 ± 102**	**65.1 ± 25.8**	**44.9 ± 20.2**	**10.8 ± 6.06**	**y = −15.3x + 183 ***	**0.92**	**Decreasing**
**TG_(55:5)**	**161 ± 137**	**92.4 ± 44**	**49.8 ± 23.4**	**14.4 ± 11**	**y = −17.9x + 216 ***	**0.98**	**Decreasing**
**TG_(55:6)**	**122 ± 143**	**100 ± 46.1**	**50.3 ± 18.8**	**8.95 ± 8.19**	**y = −14.4x + 180 ***	**0.98**	**Decreasing**
TG_(56:2)	31.6 ± 11.6	20.8 ± 10.4	30.9 ± 40	11.7 ± 4.55	y = −1.84x + 37.8	0.46	
**TG_(56:3)**	**189 ± 87.4**	**100 ± 43.6**	**93.3 ± 74.8**	**31.4 ± 14.6**	**y = −17.8x + 239 ***	**0.91**	**Decreasing**
**TG_(56:6)**	**6970 ± 6060**	**3330 ± 1460**	**1200 ± 414**	**246 ± 173**	**y = −826x + 9260 ***	**0.93**	**Decreasing**
**TG_(56:7)**	**8570 ± 7750**	**3690 ± 1650**	**1190 ± 519**	**282 ± 194**	**y = −1010x + 11,200 ***	**0.90**	**Decreasing**
TG_(56:8)	7360 ± 7600	2990 ± 1460	893 ± 372	186 ± 119	y = −875x + 9550	0.89	Decreasing
TG_(57:2)	0.611 ± 0.405	0.434 ± 0.601	1 ± 1.89	0.371 ± 0.399	y = −0.0057x + 0.648	0.00	
**TG_(57:6)**	**83.1 ± 100**	**39.6 ± 24.4**	**16.6 ± 3.26**	**5.21 ± 5.03**	**y = −9.51x + 109 ***	**0.93**	**Decreasing**
TG_(58:10)	1370 ± 1470	564 ± 268	161 ± 77.3	29.9 ± 23.1	y = −164x + 1780	0.90	Decreasing
TG_(58:7)	1160 ± 1150	484 ± 241	123 ± 65.2	20 ± 18.1	y = −140x + 1520	0.90	Decreasing
TG_(58:8)	1600 ± 1440	654 ± 354	186 ± 77.9	42.6 ± 30.7	y = −190x + 2080	0.89	Decreasing
TG_(58:9)	1730 ± 1730	730 ± 418	206 ± 106	41.2 ± 28.3	y = −207x + 2260	0.90	Decreasing
TG_(59:3)	0.283 ± 0.273	0.333 ± 0.708	0.539 ± 1.26	0.0277 ± 0.0537	y = −0.0207x + 0.454	0.12	
TG_(59:4)	0.348 ± 0.474	0.22 ± 0.449	0.372 ± 0.807	0.00429 ± 0.0121	y = −0.0326x + 0.485	0.45	
TG_(59:5)	0.212 ± 0.241	0.104 ± 0.128	ND ± ND	ND ± ND	y = −0.04x + 0.356	1.00	
TG_(59:6)	0.382 ± 0.578	0.178 ± 0.227	0.137 ± 0.174	0.108 ± 0.197	y = −0.032x + 0.446	0.81	Decreasing
TG_(59:7)	0.214 ± 0.434	0.147 ± 0.232	ND ± ND	0.0326 ± 0.0607	y = −0.0222x + 0.291	1.00	
**TG_(59:8)**	**9.09 ± 14.6**	**6.77 ± 11.9**	**5.83 ± 9.99**	**1.75 ± 2.04**	**y = −0.85x + 12.4 ***	**0.94**	**Decreasing**
TG_(60:10)	194 ± 194	73.7 ± 56.6	4.21 ± 4.15	0.838 ± 1.23	y = −24x + 252	0.86	Decreasing
**TG_(60:12)**	**79.5 ± 74.4**	**39.6 ± 28**	**13 ± 12.1**	**1.42 ± 2.02**	**y = −9.66x + 107 ***	**0.94**	**Decreasing**
**TG_(62:12)**	**63.3 ± 43.7**	**31.3 ± 17.4**	**11 ± 5.05**	**0.863 ± 0.716**	**y = −7.69x + 85.4 ***	**0.95**	**Decreasing**
**TG_(62:13)**	**4.22 ± 2.92**	**2.86 ± 3.13**	**0.546 ± 1.15**	**0.0336 ± 0.0949**	**y = −0.551x + 6.13 ***	**0.95**	**Decreasing**

**Table 2 metabolites-10-00296-t002:** This table shows the stable isotope-labelled internal standards with their ionisation products (i.e., [M+H]^+^, M+H-H_2_O]^+^, [M+Na]^+^, [M+NH_4_]^+^, [M+K]^+^, [M-H]^−^) and primary ionisation mode (positive; +ve or negative; −ve), along with their retention time (minutes). Butyryl-d7-L-carnitine (abbreviated to IS_Car_4:0-d7), N-tetradecylphosphocholine-d42 (abbreviated to IS_LPC_14:0-d42), hexadecanoyl-L-carnitine-d3 (abbreviated to IS_Car_16:0-d3), heptadecanoic-d33 acid (abbreviated to IS_FA_17:0-d33), 1,2-dimyristoyl-d54-sn-glycero-3-[phospho-L-serine] (abbreviated to IS_PS_28:0-d54), 1-palmitoyl-d31-2-oleoyl-sn-glycero-3-phosphoinositol (abbreviated to IS_PI_34:1-d31), N-palmitoyl-d31-D-erythro-sphingosylphosphorylcholine (abbreviated to IS_SM_34:1-d31), 1-palmitoyl-d31-2-oleoyl-sn-glycero-3-[phospho-rac-(1-glycerol)] (abbreviated to IS_PG_34:1-d31), 1-palmitoyl-d31-2-oleoyl-sn-glycero-3-phosphate (abbreviated to IS_PA_34:1-d31), N-palmitoyl-d31-D-erythro-sphingosine (abbreviated to IS_Cer_16:0-d31), 1-palmitoyl-d31-2-oleoyl-sn-glycero-3-phosphocholine (abbreviated to IS_PC_34:1-d31), 1-palmitoyl-d31-2-oleoyl-sn-glycero-3-phosphoethanolamine (abbreviated to IS_PE_34:1-d31), glyceryl tri(pentadecanoate-d29) (abbreviated to IS_TG_45:0-d87).

Internal Standard	Ionisation Product (*m*/*z*)	Ionisation Mode	Expected Retention Time (mins)
IS_Car_4:0-d7	239.1983	+ve	0.3
IS_LPC_14:0-d42	422.5560, 421.5498, 420.5435	+ve	0.4
IS_Car_16:0-d3	403.3610	+ve	0.5
IS_FA_17:0-d33	302.4557, 301.4495, 300.4432	−ve	1.1
IS_PS_28:0-d54	732.7741, 731.7678, 730.7615, 729.7553, 728.7490	−ve	1.4
IS_PI_34:1-d31	864.7162, 865.7225, 866.7288	−ve	2.9
IS_SM_34:1-d31	733.7632, 734.7670, 755.7451, 756.7514, 771.7190, 772.7253	+ve	3.0
IS_PG_34:1-d31	775.6939, 776.7002, 777.7065, 778.7127	−ve	3.0
IS_PA_34:1-d31	700.6509, 701.6571, 702.6634, 703.6697, 704.6760	−ve	3.4
IS_Cer_16:0-d31	548.6851, 549.6914, 550.6977, 551.7039, 566.6951, 567.7014, 568.7076, 569.7139, 590.6896, 591.6959, 606.6636, 607.6698	+ve	3.9
IS_PC_34:1-d31	790.7700, 791.7750, 812.7553, 813.7616, 828.7292, 829.7355	+ve	3.9
IS_PE_34:1-d31	747.7181, 748.7254, 749.7327, 769.7021, 770.7084, 771.7146, 785.6760, 786.6823, 787.6886	+ve	4.0
IS_TG_45:0-d87	850.2239, 851.2301, 852.2364, 853.2427, 867.2504, 868.2567, 869.2630, 870.2693, 872.2059, 873.2121, 874.2184, 875.2247, 888.1798, 889.1861, 890.1923, 891.1986	+ve	5.8

**Table 3 metabolites-10-00296-t003:** This table shows the lipid classes detected with this LC–MS lipidomics method. The number of species per lipid class and the measured adducts (protonated: [M+H]^+^, deprotonated: [M-H]^−^, protonated with water loss: [M+H-H_2_O]^+^, sodiated: [M+Na]^+^, potasiated: [M+K]^+^, ammoniated: [M+NH_4_]^+^) are also shown. The internal standard used for semi-quantification are also shown: butyryl-d7-L-carnitine (abbreviated to IS_Car_4:0-d7), N-tetradecylphosphocholine-d42 (abbreviated to IS_LPC_14:0-d42), hexadecanoyl-L-carnitine-d3 (abbreviated to IS_Car_16:0-d3), heptadecanoic-d33 acid (abbreviated to IS_FA_17:0-d33), 1,2-dimyristoyl-d54-sn-glycero-3-[phospho-L-serine] (abbreviated to IS_PS_28:0-d54), 1-palmitoyl-d31-2-oleoyl-sn-glycero-3-phosphoinositol (abbreviated to IS_PI_34:1-d31), N-palmitoyl-d31-D-erythro-sphingosylphosphorylcholine (abbreviated to IS_SM_34:1-d31), 1-palmitoyl-d31-2-oleoyl-sn-glycero-3-[phospho-rac-(1-glycerol)] (abbreviated to IS_PG_34:1-d31), 1-palmitoyl-d31-2-oleoyl-sn-glycero-3-phosphate (abbreviated to IS_PA_34:1-d31), N-palmitoyl-d31-D-erythro-sphingosine (abbreviated to IS_Cer_16:0-d31), 1-palmitoyl-d31-2-oleoyl-sn-glycero-3-phosphocholine (abbreviated to IS_PC_34:1-d31), 1-palmitoyl-d31-2-oleoyl-sn-glycero-3-phosphoethanolamine (abbreviated to IS_PE_34:1-d31), glyceryl tri(pentadecanoate-d29) (abbreviated to IS_TG_45:0-d87).

Analyte Class	No. of Species	Adducts	Internal Standard
Acyl-carnitines	48	[M+H]^+^	IS_Car_4:0-d7, IS_Car_16:0-d3
Ceramides	85	[M+H]^+^, [M+H-H_2_O]^+^	IS_Cer_16:0-d31
Cardiolipins	56	[M-H]^−^	IS_TG_45:0-d87
Diacylglycerols	6	[M+H-H_2_O]^+^, [M+Na]^+^, [M+K]^+^	IS_TG_45:0-d87
Gangliosides (GM1)	24	[M-H]^−^	IS_PG_34:1-d31
Hexosylceramides	56	[M+H]^+^, [M+H-H_2_O]^+^	IS_Cer_16:0-d31
Lyso-phosphatidylcholines	23	[M+H]^+^	IS_LPC_14:0-d42
Lyso-phosphatidyethanolamines	19	[M+H]^+^	IS_LPC_14:0-d42
Lyso-phosphatidylinositols	19	[M-H]^−^	IS_PI_34:1-d31
Lyso-phosphoserines	20	[M-H]^−^	IS_PS_28:0-d54
Lyso-cardiolipins	23	[M-H]^−^	IS_TG_45:0-d87
Monoacylglycerols	1	[M+H-H_2_O]^+^, [M+Na]^+^, [M+K]^+^	IS_TG_45:0-d87
Phosphatidic acids	26	[M-H]^−^	IS_PA_34:1-d31
Phosphatidylcholines	43	[M+H]^+^	IS_PC_34:1-d31
Phosphatidylethanolamines	19	[M+H]^+^	IS_PE_34:1-d31
Phosphatidylglycerol	34	[M-H]^−^	IS_PG_34:1-d31
Phosphatidylinositols	21	[M-H]^−^	IS_PI_34:1-d31
Phosphatidylserines	36	[M-H]^−^	IS_PS_28:0-d54
Sulfatides	72	[M-H]^−^	IS_PG_34:1-d31
Sphingomyelins	54	[M+H]^+^, [M+Na]^+^, [M+K]^+^	IS_SM_34:1-d31
Triacylglycerides	89	[M+H]^+^, [M+NH_4_]^+^, [M+Na]^+^, [M+K]^+^	IS_TG_45:0-d87

**Table 4 metabolites-10-00296-t004:** This table shows the dietary fat composition of each of the four experimental diets fed to Sprague–Dawley rats (n = 8–9 per group). MCT: medium chain triglyceride oil.

Diet	Corn Oil	MCT Oil	Beef Tallow
1	50%	16.4%	3.6%
2	35%	28.7%	6.3%
3	20%	41.0%	9.0%
4	5%	53.3%	11.7%

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
