# Peer review of "LC–MS Lipidomics: Exploiting a Simple High-Throughput Method for the Comprehensive Extraction of Lipids in a Ruminant Fat Dose-Response Study"

_metabolites, 2020, doi:10.3390/metabo10070296_

Round 1

Reviewer 1 Report

The manuscript introduces a new high-troughput tissue sample protocol for lipidome analysis. The authors have shown that the proposed method outperforms common Folch protocol, and provide required statistical analysis. The method is applied to a lipidomic study and shows promising results for untargeted lipidomics.
However, I have some questions and comments:

Major comment:

1) Authors are rather careless with the paper size. Table and figure captions are unreadable. Table 1 should be moved to supplementary. Why do athors waste 1.5 pages (Chemicals & standards) saying same words "purchased from Sigma" and give order numbers? This all should be moved to supplementary

Minor:

1. How was the precipitation agent's composition chosen? Was any experimental optimization performed, or are there any heuristic justification for the choice?
2. It is not clear from the text, to what kind of samples does figure 1 refer to? Were two protocols compared on tissue samples? If yes, then what are the expectations from the proposed method when it is applied on liquid sample (urine, blood) analysis?
3. The comparison of two methods was done with spiked mixture of internal standards. However in the case of tissues the data for sample lipids should be additionally given, to consider for example the effects on the cell membrane extraction. It could be the total number of detected lipids of the same sample after preparation by two protocols, or the corresponding intensities.

I suggest considerable recomposing of the paper and re-review

Author Response

Reviewer 1’s comments/questions: This reviewer highlighted 7 points that needed to be addressed; the reviewers comment/question (in highlighted/bold) and the authors response are shown below each comment/question.

Comment/question 1: Authors are rather careless with the paper size.

Replies/answer 1: The length of the manuscript is necessary to include all the detail to reproduce both the method and the results, omitting any of the details would detract from the manuscript as a whole. However, we have made every effort to reduce the size by re-writing sections, grouping/consolidating information and utilising references.

Comment/question 2: Table and figure captions are unreadable.

Replies/answer 2: We have made considerable improvements to the captions of both the tables and figures in this manuscript by reducing the size and fragmentation of the text.

Comment/question 3: Table 1 should be moved to supplementary.

Replies/answer 3: Table 1 contains the key results of the manuscript and therefore we have kept this in the main body of the manuscript. It would not be appropriate to move this table to the supplementary material since it is not complementary to the manuscript but an essential part of it. We feel that even shortening this table would be disingenuous since the reporting of negative results (i.e. results that are not statistically significant) are just as important as reporting the statistically significant results and would introduce an unwanted bias. The results presented in table 1, both the significant and non-significant ones, are important in understanding the relationship of ruminant fat and odd chain lipids in vivo, as well as demonstrate the full capabilities of the method.

Comment/question 4: Why do authors waste 1.5 pages (Chemicals & standards) saying same words "purchased from Sigma" and give order numbers?

Replies/answer 4: We have significantly decreased the size of the chemicals & standards section by grouping each standard by the provider. However, we have kept the full details of the standards used along with their order numbers. As stated in the ‘Author guidelines’, the authors must describe with sufficient details everything so that the methods and results can be replicated. By keeping the order numbers of the standards it allows the reader to fully understand each chemical & standard and look into their individual purities and compositions.

Comment/question 5: How was the precipitation agent's composition chosen? Was any experimental optimization performed, or are there any heuristic justification for the choice?

Replies/answer 5: The new extraction method was produced as a result of two situations: firstly a collaborator requesting full lipidomics analysis of their samples but including relatively polar acyl‑carnitines and glycosphingolipid and secondly we needed an extraction method that allowed a single-extraction to be optimal whilst being high-throughput, cost-effective, use solvents that would dry quickly at a low temperature and cause the protein pellet to collect at the bottom of the extraction tube (allowing for high-throughput and automation). When looking at suitable solvents based on their relative polarity (especially dielectric constant), the solubility of salts and the boiling points of organic solvents, as well as H&S considerations and the availability & cost of solvents, acetone was the chosen candidate. I have done a significant number of comparisons on different extraction solvent systems, where chloroform: methanol (2:1) produces the best protein precipitation. By combining acetone with the chloroform: methanol (2:1) it has an overall improved lipid extraction capability whilst meeting all the requirements we needed. This background information is outside of the scope of the text of the manuscript.

Comment/question 6: It is not clear from the text, to what kind of samples does figure 1 refer to? Were two protocols compared on tissue samples? If yes, then what are the expectations from the proposed method when it is applied on liquid sample (urine, blood) analysis?

Replies/answer 6: We have changed the table and now made clear the caption to figure 1 so that it is explicitly refers to the intensities from the rat liver samples. Although we have a lot of data for this extraction method on fluid samples (plasma, serum and lysates) we have provided the details for the liver samples in this manuscript. However, since plasma/serum samples contain a large amount of polar-lipids, the performance of the protein precipitation liquid extraction (chloroform: methanol: acetone, ~7: 3: 4) over the Folch method is even more pronounced.

Comment/question 7: The comparison of two methods was done with spiked mixture of internal standards. However in the case of tissues the data for sample lipids should be additionally given …. It could be the total number of detected lipids of the same sample after preparation by two protocols, or the corresponding intensities.

Replies/answer 7: We have now added in a figure and a table into the supplementary material (see supplementary figure S1. & table S1) that specifically shows the comparison of the two extraction methods described on the extraction of endogenous lipids. The protein precipitation liquid extraction (chloroform: methanol: acetone, ~7: 3: 4) out performs the Folch method both in the intensity of the individual lipid classes and the total number of lipid detected for each method.

Reviewer 2 Report

Jenkins et al describe an interesting workflow how to measure lipids using a single phase extraction with a mixture of chloroform: methanol: acetone (7: 3: 4). They compare this extraction method to one of the standard lipidomics workflows (Folch method using chloroform:methanol:water) and apply the new extraction method to study ruminant fat dose-response. The paper is well written, the analytical workflow described in detail and an interesting application is shown. After considering minor comments, I recommend to publish the article in Metabolites.

  • How were the ruminant samples derived? Are there any ethical considerations? Please add the full information at least in the supplementary
  • Extraction method: The authors refer to the workflow chloroform: methanol: acetone as novel extraction strategy and protein precipitation single phase extraction, however similar extraction strategies were already applied earlier such as for thin layer chromatography in the 1970s (e.g. the Lipid Handbook, F. Gunstone, separation and isolation procedures). Please also cite earlier works… The new application in the field of lipidomics is still interesting due to the automation potential
  • The authors claim that other biphasic methods are not able to properly extract (due to partition in the two phases) more polar lipid classes such as acyl-carnitines and gangliosides, which the chloroform: methanol: acetone can do more quantitatively. However, especially for these classes the presented data is not conclusive: All carnitines elute at the void volume or close to it. The gangliosides are not properly assigned (what kind of exact classes are detected? GM1, GM2, GM3, etc). If the authors can only see GM3s, what is the improvement to other biphasic systems?
  • Figure 1: Except for the class of carnitines and PS, I do not see a significant extraction improvement (only a slight trend towards better extraction) compared to Folch. And as carnitines elute at the void, this significance could be hampered
  • The authors state that IS_TG_45:0-d87 has a +H adduct. Using the presented LC-MS with the IPA gradient and Orbitrap detection mostly M+NH4 should be there (as also correctly shown in Figure 5) and such adducts usually do not occur.
  • Data processing: How were the different lipid species assigned? Only in targeted manner or also by using non-targeted workflows such as Lipid Search? If yes, what kind of filtering was used?

Author Response

Reviewer 2’s comments/questions: This reviewer highlighted 8 points that needed to be addressed; the reviewers comment/question (in highlighted/bold) and the authors response are shown below each comment/question.

Comment/question 1: How were the ruminant samples derived? Are there any ethical considerations?

Replies/answer 1: The ruminant samples were ethically derived from Bio-Serv (Flemington, New Jersey, U.S.A.), these details are within the original manuscript from where these samples originally derived which is also appropriately cited within the text of this manuscript. We have not included these details within this manuscript to avoid elongating the text any further.

Comment/question 2: The authors refer to the workflow chloroform: methanol: acetone as novel extraction strategy and protein precipitation single phase extraction, however similar extraction strategies were already applied earlier such as for thin layer chromatography in the 1970s (e.g. the Lipid Handbook, F. Gunstone, separation and isolation procedures).

Replies/answer 2: As the reviewer correctly points out there are many examples in the literature of protein precipitation methods utilising different solvent systems, however, to our knowledge and after a considerable amount of literature search the method described in this manuscript is novel.

Comment/question 3: The authors claim that other biphasic methods are not able to properly extract (due to partition in the two phases) more polar lipid classes such as acyl-carnitines and gangliosides, which the chloroform: methanol: acetone can do more quantitatively. However, especially for these classes the presented data is not conclusive.

Replies/answer 3: A comparison of the two extraction methods is shown in figure 1 where the majority of the lipid internal standards are better extracted with the protein precipitation liquid extraction (chloroform: methanol: acetone, ~7: 3: 4) than the Folch liquid-liquid extraction (please bear in mind the y-axis is logarithmic so difference may look less pronounced). We have now included a figure and a table into the supplementary material (see supplementary figure S1. & table S1) that specifically shows the comparison of the two extraction methods described on the extraction of endogenous lipids. The protein precipitation liquid extraction (chloroform: methanol: acetone, ~7: 3: 4) out performs the Folch method both in the intensity of the lipid classes and the total number of lipid detected for each method.

Comment/question 4: All carnitines elute at the void volume or close to it.

Replies/answer 4: Since the LC-MS analytical method is being used for full lipidomics (~800 analytes) there had to be some compromises. Taking into account the instrument limitations, the entire range of lipids, the different chemistries, etc. then some analytes will not have optimum analytical conditions. The acyl-carnitines do elute close to the solvent front, however, there is no insource fragmentation resulting in co-eluting isomers, we have found their quantitation to be robust and not significantly affected by sample differences. Furthermore, the extraction method was specifically designed to avoid the extraction of salts from within the samples, therefore, analytes eluting close to the solvent front are less affected by sample differences (assuming similar sample sizes/volumes are used); this is particularly evident with biological circulatory fluids (plasma/serum).

Comment/question 5: The gangliosides are not properly assigned (what kind of exact classes are detected? GM1, GM2, GM3, etc). If the authors can only see GM3s, what is the improvement to other biphasic systems?

Replies/answer 5: We have now changed the names of the gangliosides to the GM1 nomenclature as suggested. The improvements of this extraction method over the Folch liquid-liquid extraction method are shown in figure 1 in the manuscript and also in supplementary figure S1. & table S1.

Comment/question 6: Figure 1: Except for the class of carnitines and PS, I do not see a significant extraction improvement (only a slight trend towards better extraction) compared to Folch.

Replies/answer 6: A comparison of the two extraction methods is shown in figure 1 where the majority of the lipid internal standards are better extracted with the protein precipitation liquid extraction (chloroform: methanol: acetone, ~7: 3: 4) than the Folch liquid-liquid extraction (between ~30% to ~2500% higher for seven of the internal standards) (please bear in mind the y-axis is logarithmic so difference may look less pronounced). We have now included a figure and a table into the supplementary material (see supplementary figure S1. & table S1) that specifically shows the comparison of the two extraction methods described on the extraction of endogenous lipids. The protein precipitation liquid extraction (chloroform: methanol: acetone, ~7: 3: 4) out performs the Folch method both in the intensity of the lipid classes and the total number of lipid detected for each method.

Comment/question 7: The authors state that IS_TG_45:0-d87 has a +H adduct. Using the presented LC-MS with the IPA gradient and Orbitrap detection mostly M+NH4 should be there (as also correctly shown in Figure 5) and such adducts usually do not occur.

Replies/answer 7: In the manuscript we have already described that the IS_TG_45:0-d87 internal standard has multiple adducts: [M+H]+, [M+NH4]+, [M+Na]+, [M+K]+. Using the orbitrap and the Xcalibur software we are able to produce an extracted ion chromatogram that includes all of the different adducts into a single chromatogram. As the reviewer correctly states, the ammoniated adduct is usually the most abundant (as shown in figure 5. in the manuscript), however, to maximise the intensity we include all of the different adducts described. Although the protonated IS_TG_45:0-d87 is almost inconsequentially low in comparison to the other adduct, it is still detectable and therefore is included.

Comment/question 8: Data processing: How were the different lipid species assigned? Only in targeted manner or also by using non-targeted workflows such as Lipid Search? If yes, what kind of filtering was used?

Replies/answer 8: The targeted data processing method is already described in the manuscript where a peak in the extracted ion chromatogram at the expected retention time (see supplementary table S2.) is integrated thus producing the area intensity for that m/z. This analyte area intensity is then divided by the corresponding internal standard area intensity to produce an area ratio. This area ratio is then used to produce the semi-quantitative concentrations for that individual lipid.

Round 2

Reviewer 1 Report

I recommend to accept the manuscript

Reviewer 2 Report

The authors have addressed all comments. The additional information on the extraction comparison with chloroform/ethanol/water mixtures (Table S1 combined with Figure S1) is very valuable. Overall, I recommend to publish the manuscript in Metabolites